# Biomolecules and Electrochemical Tools in Chronic Non-Communicable Disease Surveillance: A Systematic Review

**DOI:** 10.3390/bios10090121

**Published:** 2020-09-10

**Authors:** Ana Lúcia Morais, Patrícia Rijo, María Belén Batanero Hernán, Marisa Nicolai

**Affiliations:** 1CBIOS—Universidade Lusófona Research Centre for Biosciences & Health Technologies, Campo Grande 376, 1749-024 Lisbon, Portugal; ana.nunes@edu.uah.es (A.L.M.); patricia.rijo@ulusofona.pt (P.R.); 2Department of Biomedical Sciences, Faculty of Pharmacy, University of Alcalá, Ctra. A2, Km 33.600–Campus Universitario, 28871 Alcalá de Henares, Spain; 3iMed.ULisboa-Research Institute for Medicines and Pharmaceutical Sciences, Universidade de Lisboa—Faculdade de Farmácia, Av. Prof. Gama Pinto, 1649-003 Lisbon, Portugal; 4Department of Organic & Inorganic Chemistry, Faculty of Pharmacy, University of Alcalá, 28805 Madrid, Spain

**Keywords:** non-communicable diseases, electrochemistry, hydrogen peroxide, dopamine, uric acid, ascorbic acid, reactive oxygen species, antioxidants

## Abstract

Over recent three decades, the electrochemical techniques have become widely used in biological identification and detection, because it presents optimum features for efficient and sensitive molecular detection of organic compounds, being able to trace quantities with a minimum of reagents and sample manipulation. Given these special features, electrochemical techniques are regularly exploited in disease diagnosis and monitoring. Specifically, amperometric electrochemical analysis has proven to be quite suitable for the detection of physiological biomarkers in monitoring health conditions, as well as toward the control of reactive oxygen species released in the course of oxidative burst during inflammatory events. Besides, electrochemical detection techniques involve a simple and swift assessment that provides a low detection-limit for most of the molecules enclosed biological fluids and related to non-transmittable morbidities.

## 1. Introduction

In the past decades, a significant increase in chronic non-communicable diseases (NCDs) has been observed in individuals of all ages, making it one of the main leading causes of worldwide death [1]. These NCDs are chronic and non-transferable health conditions; closely related to an individual’s lifestyle and it’s contact with environmental pollutants and many emerge from inappropriate diets and detrimental behavioural habits, such as tobacco and alcohol use [2].

As living organisms are in dynamic equilibrium with a direct vicinity environment, any metabolic imbalance may disrupt other related parameters such as glucose intolerance, insulin resistance, high blood pressure, among others, in a domino effect. For instance, metabolic syndrome derived from noxious dietary habits, is responsible for the development of tissues inflammation that instigates human organism oxidative stress. In turn, at overwhelming levels; reactive oxygen species (ROS) affects several components of insulin-receptor signal transduction, instigating human body insulin resistance. This lack of sensitivity to insulin action stimulates the appearance of type-2 diabetes *mellitus* due to the permanent high level of glucose in blood [3]. As such, every single medical condition is strongly related, and it is enrolled in the fluctuation of several biological factors. For example, biological factors such as stress can activate the hormonal system, increase blood pressure and reduce immune response, or the consumption of foods high in saturated fats can be related to atherosclerosis progression and increase the risk of a heart attack. The control and surveillance of the amount of such biological parameters (biomolecules) in the patient’s fluids, is thus of significant importance for precise, correct, and relevant treatment.

At present, this surveillance has often been carried out by several electrochemical techniques that through the survey of the redox reactions between electroactive biologicals elements and an electrode surface, allows the accurate quantification of these biological species [4,5], as the regularly used amperometric glucose biosensors, for blood sugar supervision.

This evaluation requires the application of high conductive electrode material to give a considerable electrochemical signal as a response when the electroactive biomolecules are present. Therefore, it allows one to attain a high sensibility towards the targeted biological metabolites [2], by ensuring the quantification of specific elements in trace amounts within organic fluids. Simultaneously such electrode material should be designedly gathered in order to produce a single electrochemical response from the target biomolecule (such as the modified electrodes), without experiencing possible foreign interferences from biological species coexisting in biological fluid-samples. Such a procedure allows the screening of biological metabolites with high selectivity.

In the present study, the electrochemical techniques were selected for the surveillance of common NCD because it owns some special features that turn it one of the most suitable facilities for an early and accurate vigilance of health disorders. In particular, the capacity to detect trace amounts of biological species, their miniaturisation that offers the possibility to access convoluted areas of the human organism, and their portability, which makes it easier to use in clinical applications.

In general, electrochemical appliances provide many advantages over conventional analytical systems, like the ability to reproducibly handle minimal amounts of samples, low reagent consumption, reduced processing time, ease of proceeding and low-cost analysis.

The substantial grow of chronic non-communicable diseases amongst the less elderly population is increasingly a concern worldwide. It is therefore essential to combat and prevent the progression of this kind of long-lasting health conditions, by defining monitoring strategies. These might be achieved through the applications of reliable and efficient techniques and methodologies that allow an accurate early detection, leading to proficient clinical outcomes that promote general public health.

Present work aims to review concepts related to most common NCD; biosensors; Electrochemical techniques employed on disease diagnosis; biological recognition elements for electrochemical sensing; Reactive species sources and role in physiological and pathological processes; Human body antioxidant mechanisms.

This overview sustains the development of tools with specific characteristics in order to trade them for use in biomolecule analysis and early detection of NCD.

## 2. Review Survey Methodology

For the assembly of this review paper a systematic literature survey was conducted in accordance with PRISMA guidelines [6]. This scrutiny was made on PubMed database.

Non-communicable diseases recent information was collected and selected, based on several search items: (Non communicable diseases OR NCD), AND (reactive oxygen species OR ROS); Electrochemical Biosensors, AND (Non communicable diseases OR NCD); Non-enzymatic biosensor AND (Non communicable diseases OR NCD); Amperometric biosensors AND (Non communicable diseases OR NCD); Comorbidity AND (Non communicable diseases OR NCD); Oxidative stress AND (Non communicable diseases OR NCD); Signalling messenger AND (Non communicable diseases OR NCD), separately combined. The research has been limited to English language reports over the past 30 years, including in vitro, and population health survey studies. All the abstracts initially gathered, were screening and evaluated according to their solid closing remarks.

To review the literature on the leading factors that trigger the major non-communicable diseases, and their mutual relations, along with the influence of abnormal levels of certain biomolecules on the progression of each of those non-communicable diseases, it was used the following search terms: (hydrogen peroxide), (uric acid), (ascorbic acid OR vitamin C), (dopamine), each item separately combined with (obesity OR metabolic syndrome), (type 2 diabetes *mellitus*), (cardiovascular diseases), (chronic obstructive respiratory disease), (neurodegenerative diseases OR neurological disorders) and (cancer). The investigation has been limited to the past 20 years, in English language, including in vitro, population health survey or animal studies. The final survey was held in June 2020. The titles and abstracts of spotted articles through the initial search were considered, and the most meaningful articles were afterwards extensively analysed. Whenever the study results remained somehow inconclusive or inferring a wavering conclusion, the initial deemed articles were excluded.

## 3. Results

From the total 47,706 articles obtained through the database used, along with the supplementary 119 articles retrieved from other sources, (among which Science Direct database), 855 articles were selected. These latter works were analysed in the light of the evidence provided on broader populations, and all the researches resulting from studies with unrepresentative groups were discarded.

Figure 1 shows the flowchart with the results obtained from the above methodology. From 855 records screened, 574 articles were assessed for eligibility, 396 excluded, and the remainder were included in qualitative analysis.

Such a protocol process applied in the review analysis has resulted in the following twelve sections described below.

## 4. Non-Communicable Diseases

Non-communicable diseases (NCDs) are generally long term and slow-progression chronic health disorders, non-infections, and directly non-transmissible amongst individuals. At present, the high prevalence of chronic NCDs makes it known as the commonest cause of disability in the modern world, and the major threat to public health, being now responsible for 71% of annual deaths worldwide [1].

Among the total spectrum of non-transmittable health disorders, six main prevailing diseases are affecting the world’s population: obesity, type 2 diabetes *mellitus* (T2DM), cardiovascular (CV) diseases, chronic obstructive pulmonary (COP) disease, neurological diseases, and cancer [7].

As the risk factors for NCDs emerge mainly from the way of life, customs, and behavioural habits, such as poor dietary practices, physical inactivity, smoking, and excessive alcohol consumption, these are commonly mentioned as lifestyle diseases. Accordingly, inappropriate lifestyles conduct to metabolic disorders occurrence and, significant changes in an individual’s physical conditions, which precede some of the NCDs.

In some cases, several chronic health conditions are concurrently present in the same individual, which leads to a state of multimorbidity. Most often, there is an interrelation between those multiple health disorders, such as obesity, T2DM, and CV problems. An improper diet associated with sedentary habit lifestyle conducts to metabolic syndrome with the unbalance of nutrients for a correct function of the body. Mainly these bad habits result in excess blood glucose and other carbohydrates. With time, such an overload of carbohydrates worsens the tissue response to insulin, so not lowering the glucose levels within the blood. This, in turn, leads to the development of another metabolic syndrome, the T2DM that may trigger some CV disorders like atherosclerosis [8,9].

Concretely, obesity is a chronic metabolic disease that affects a substantial portion of the population worldwide. Higher-than-average poor-diet consumption leads to metabolic syndrome with adverse health consequences. For instance, long-term carbohydrates excessive ingestion stimulates the pathological condition where cells no longer react to insulin presence, triggering the T2DM metabolic syndrome. Besides, overweight individuals exhibit an increase of perivascular adipose tissue, which increases their vascular resistance to the blood flow, and consequently the effort of the heart to pump blood throughout the body, which can lead to several CV disorders [10].

On its own, obesity is directly associated with adverse health consequences and is the major risk factor to other chronic diseases, such as T2DM, coronary artery (CA) disease, stroke, cardiac arrhythmias, non-alcoholic fatty liver disease, asthma, obstructive sleep apnoea and cancer [11].

Ultimately, an insulin resistance along with blood-glucose management pancreatic islet cell malfunction makes unavoidable the development of T2DM [12].

As a complex highly developed living organism, the human body [13] is in constant dynamic homeostatic balance with its surroundings. As a result, there is a permanent exchange of biological elements with its vicinity to preserve the vital physiological parameters. Therefore, a healthy organism is dependent on the regulation of physiological parameters, as well as the amount of some biomolecules found in organism fluids.

In general, such parameters involve other specific properties of the human organism, namely the body temperature, fluids pressure, and physiological level of several biomolecules. 

Under physiological conditions, various vital parameters are within specified levels, fundamental for the normal function of the overall complex living organism [14].

When the human organism experiences a shift in redox balance towards oxidized molecules and the generation of reactive oxygen species (ROS), as opposed to the organism’s ability to depleting these reactant intermediates, a state of oxidative stress occurs.

Whenever biological system (cells, tissues, and organs), undergoes a disruption of its normal physiological status, it can adapt reversibly as a response to adverse conditions, to maintain a steady state (also called homeostasis). Accordingly, cells are temporarily stimulated to proceed to adapt to the new demands of the organism. For instance, in overburden conditions, cells are stimulated to enlarge due to their increased performance demand, as in the case of high blood pressure, with the cardiac muscle cells being required to increase work of pumping blood demanded [15].

Over severe or long-termed stress conditions, cellular adaptation response may be overwhelmed and trigger abnormal mutations on cells and tissue biology. This spoilage might lead to other injuries, like the accumulation of neoplasia tissue, aging, or dead tissue, which normally results in cancer occurrence [16].

Owing to such oxidative stress, there is ROS-mediated damage to important organelles and biomolecules of the cellular metabolic system. This excessive exposure of the organism to ROS will disturb the redox homeostasis, and therefore induce several deleterious effects on the living cells, such as membrane cell damage, and impairment of cellular functions and enzyme activity, leading to the emergence of some chronic/degenerative diseases such as diabetes, cancer, CV diseases, Parkinson’s, Alzheimer’s, among others [17].

Under stress conditions, eukaryotic living organisms testify a significant increase in oxidative species among organ constituent cells, tissues, and fluids. These species react extremely fast with organic molecules, altering the structure and functional capability of phospholipids, nucleic acids, and proteins, etc. Such an amendment will affect the functionality of cell membranes’, enzymatic activities, and gene expression, which may result in organism oxidative serious damage [9,18].

Pathological conditions can arise from a dysregulation on the oxidative stress level within a living organism, with the development of molecular damage, in a sequence mode with a ripple effect [9].

Depending on the affected biological system, there are several fundamental molecules involved in the oxidative damage and directly associated with the various chronic no communicable disorders. The detection and quantification of those relevant metabolites is the necessary route for the prevention of further complications associated (related disorders) and the application of highly effective therapies. Therefore, the early detection of key metabolites involved in the oxidative damage developed during physiological disorder is fundamental for the precise knowledge of the injury progression state [19].

Nowadays, the major causes of death and concern noteworthy health disorders are CV diseases, neurodegenerative disorders, chronic respiratory injuries, and cancer [5].

These injuries are deeply related to the unregulated levels of certain metabolites and molecular compounds in the body. Such biological compounds make part of the body’s metabolic system, as precursors of other species essential to the functioning of human organs, or as regulatory agents and/or extra and intracellular messengers. For instance, irregular levels of hydrogen peroxide (H_2_O_2_) in blood or urine may be an asymptomatic indication of an electrolyte body fluids dysregulation, which actively contribute to the constriction of an endothelial blood vessel with high CV disorder risk [20].

Likewise, the detection of low levels of dopamine content in cerebrospinal fluid or blood might be a signal of lower activity of nitric oxide (NO) brain-signalling molecule, which regulates cerebral blood flow. This may also be associated with hypothyroidism syndrome or Alzheimer’s disease [10].

The maintenance of all organism systems redox balance along with all metabolites involved in NCDs syndromes is therefore of paramount importance.

## 5. Biosensors

Taking into consideration the efficient surveillance and treatment of pathologies in individuals, a comprehensive evaluation of their physiological state is fundamental [4]. For this purpose, an accurate qualitative and quantitative inquiry of their physiological attributes, such as the levels of physical-chemical, metabolic, and biomolecular parameters, is essential.

Through a physiological assessment of specific biological metabolites associated with an individual’s health status, it is possible to have an insight of its organism’s condition and functioning, thus facilitating an early diagnosis in a possible pathological process.

Based on precise knowledge of the referred parameters, it is possible to prevent and control the development of diseases such as tumours, cardiac and coronary heath diseases, or metabolic disorder, to contain its spread, as well as to avoid the proliferation of related disorders [21].

Alagappan et al. [22] had developed a high-performance cholesterol biosensor through the co-immobilization of a metal-carbon-polymer nanocomposite with cholesterol oxidase enzyme (Au-f-MWCNT-PPy-ChOx), with high selectivity, sensitivity and reproducibility over a ten-fold linear range under pH 7.0 detection condition. Since in higher levels, cholesterol is strongly associated with arteriosclerosis development, which in turn it is the dominant cause of several severe life-threatening diseases like peripheral vascular disease, myocardial infarction, cardiovascular disorders and stroke [23,24], it is essential to control the cholesterol levels in biological fluids.

The in-depth knowledge of human body conditions’ is also a considerable advantage to apply an effective therapy in the treatment and potential cure [13].

Accordingly, a careful qualitative and quantitative assessment of measurable physiological parameters is crucial, including several physical properties and/or biological compounds existing in the human body [13].

The evaluated physical parameters of human physiology involve blood pressure, body temperature, blood flow rate, blood viscosity, and the electromagnetic field of the organism [13].

Measurable chemicals and biochemical pertain to the detection of biological substances and their concentration in body fluid, such as ions, microorganisms, proteins, or other fundamental molecules like lipoproteins, and unsaturated fatty acids [25]. Therefore, the detection of different protein complexes such as the enzymes, nucleic acids, antibodies, and antigens is also possible [26]. This approach is accomplished through sensitive and selective detection analytical tools, such as biosensors.

Analytical research conducted by biosensors began in 1962 [27] with the introduction of Clark and Lyons’ glucose oxidase sensor, in the human body. Since then numerous useful applications involving biosensors, have been introduced and commercialized [28].

In practice, biosensors are self-contained analytical devices based on the recognition of biological elements [17]. This device comprises a bio-recognition element in which the target biological component being analysed is disposed of, directly connected to a physical-chemical element (a transducer) through an interface. In turn, this transducer generates measurable electrical signals that correspond to the quantitative and semi-quantitative species-specific analytical information of the analysed sample.

Biosensors are applied in different situations depending on the target species to be analysed and are categorised according to the bio-recognition element used, and the type of physical-chemical characteristic observed [29].

Therefore, biosensors may involve transfers of heat, light, electric current, mass exchange, or pH, etc., depending on the physical-chemical property involved in the received signal transduction, in accordance to the biological receptor [19]:ElectricalChemical (electrochemical/impedance biosensor)Optics (optical/surface plasmon resonance biosensor)Thermal (enzymatic thermo-biosensor)Mechanical: Piezoelectric (quartz crystal microbalance biosensor).

Every transductor has its specificity, generating different electrical magnitudes range, according to their specific nature [19]. Thus, the correct selection of a type of biosensor for the analysis of a biological sample relies on the bio-recognition principle, which is in twofold broad categories [30]:Catalytic biorecognition: enzymes, other macromolecules;Biorecognition by affinity: antibodies, nucleic acids, microorganisms.

The electrical signals emerging from the transducer are then scaled-up and processed for subsequent visualization of the resulting data.

## 6. Electrochemical Methods Employed on Disease Diagnosis

Recently, electrochemical biosensors detach among analytical techniques, as the most powerful for the evaluation of biologic material, in clinical surveys [4]. Commercial electrochemical biosensors present some promising practical features against other analytical techniques that turn it suitable in clinical diagnosis surveys. The speed of response, simple operation procedure, and direct analyte detection, as well as their ability for reuse, low cost-effective manufacture, miniaturization feasibility, and high portability, makes it viable techniques for the detection of a real-time sample [31]. As a result, over the last few years, numerous electrochemical biosensors have been developed and widely used in early detection of various health disabilities based on accurate monitoring of specific biological molecules, also called biomarkers [4].

This physiological surveillance at the molecular level is accomplished by the quantitative evaluation of some compounds (organic and/or inorganic), metabolites, and other molecules present in the human body.

Analysed bio-components (bio-recognition elements of a biosensor) are electrically responsive compounds that react to a disturbance of electric nature, when in contact with an electrochemical transducer, thus altering its oxidation state [4]. Those compounds usually consist of macromolecules, such as enzymes, antibodies, antigens, tissues, living cells, and electrolyte ions, ongoing present in various bodily fluids. 

Accordingly, biosensing detection is peculiarly valuable in the quantitative analysis of chemical or biochemical species, including genetic material, blood, urine, serum, saliva, sweat, and other bodily fluids. Therefore, regular samples can be systematically evaluated by non-invasive and straightforward procedures, only requiring minimal intervention for the collection of pertinent fluids [32,33].

Chemical information obtained from electrochemical biosensors, enables the precise control of some important endogenous species, for instance, several signalling molecules, responsible for the broadcast of important information between cells in the human organism. This renders electrochemical biosensors attractive devices for the rapid and accurate evaluation of some specific biomarkers directly related to the individual’s state of health [17].

Different interfacial electrochemical techniques are applied in the biosensing procedure (stated in Table 1), depending on the required information. The established parameters will then relate to other physical and chemical quantities.

Therefore, electrochemical biosensors can be categorised according to the operating principle involved [34]:

Alternatively, biosensors’ can be either ranked in accordance to the biorecognition-element used [19]:Enzyme-based electrochemical biosensors;Whole-cell electrochemical biosensors;Bacteriophage-based electrochemical biosensor;Electrochemical nucleic acids biosensor;Aptamer-based electrochemical biosensors;Electrochemical immunosensors.

For the sake of plainness, this work will focus only on certain electrochemical techniques mentioned in Table 1: potentiometry, amperometry/voltammetry, and impedimetry techniques.

The potentiometric quantitative technique relies on the measurement of a potential difference between the working electrode that reacts/responds to the analytes’ ionic activity, and the fixed and known potential from the reference electrode, which is free from interferences.

A potentiometric biosensor provides information about the ion activity in an electrochemical reaction, which will subsequently co-relate with the analyte concentration through a logarithmic function. Thus, the analytical signal generated by a potentiometric sensor corresponds to the variation in the concentration of ionic species.

The electrochemical reaction occurs in an unchanged species concentration setup, according to Figure 2. 

Potentiometric measurements are ruled by the Nernst equation principles [17] that relate the potential difference between two electrodes in an electrochemical cell (the reference and working, electrodes), with the ionic activity of the analyte species, as shown in Equation (1), for the half-cell reaction
(1)$$a\;\mathrm{Ox}+n_{i}{\;e}^{-}\to \;b\;\mathrm{Red}.\;{E\;=\;E}^{0}-\frac{\mathrm{RT}}{n_{i\;}F}\ln\left(\frac{{{[a}_{i}{}^{\mathrm{reduced}}]}^{b}}{{\;[a}_{i}{}^{\mathrm{oxidised}}{]}^{a}}\right)$$
where, $E^{0}$ is the standard electrode cell potential of the sensor electrode, $a_{i}{}^{\mathrm{reduced}}$, and $a_{i}{}^{\mathrm{oxidized}}$, concerning the activity of the reduced and oxidized ions, measured by the working electrode, according to the reference electrode in the sensor, respectively, *R* is the universal gas constant (8.31451 J K^−1^ mol^−1^), *T* is the absolute temperature (K), *F* is the Faraday constant (F = 96.485 C mol^−1^), and $n_{i}$, corresponds to the number of electrons involved in the elementary redox reaction. 

According to the specificities of target molecules, there are several types of potentiometric biosensors. Indeed, such electrochemical measurements have provided the evaluation of several vital biomolecules, such as organic and inorganic species, like sugars [35], urea [36], antibiotics [37], neurotransmitters [38], environment pollutants [39], carbon dioxide [26] and many ionic species, like salts and minerals [17,26]. Therefore, it is regularly used to determine the analytical concentration of biologically relevant electrolytes in physiological fluids, such as Na^+^, K^+^, Ca^2+^, and Cl^−^, which are responsible amongst other functions, for conduct electrical impulses in the body [17,40].

Recently Urbanowicz et al. [41], reported on a very small multiple biosensing platform of 10 mm diameter, relying on solid contact ion-selective electrodes for the detection of electrolytes Na^+^, K^+^, Ca^2+^, Mg^2+^, and Cl^−^ in fresh human saliva samples. This electrolyte evaluation is essential to manage the progression of a number of non-communicable diseases, such as type-2 diabetes mellitus and lipid profile [42].

The quantification through amperometry technique is accomplished by the measurement of the electric current under a controlled value applied constant potential, as a function of another experimental variable (e.g., time, reagent concentration, added reagent volume). When amperometric measurements accomplish is followed in an unstirred sample solution, the mass transport of the active species onto the surface of the electrode, where redox reaction occurs, is exclusively driven by diffusion. Such a diffusion-limited current process is proportional to the concentration of the analysed electroactive species, in the analyte solution [43]. Under these conditions, amperometric biosensors provide information on the concentration of the analyte.

Amperometric detection requires a three-electrode set-up, where the potential difference is applied to the working electrode concerning the reference electrode. In contract, the faradaic electric current that resulted from the electrochemical reaction occurring at the surface of the working electrode is registered in the course of the electrons flow between, working and auxiliary electrodes [44,45,46]. This process is schematically represented by Figure 3.

In many aspects, voltammetry is identical to amperometry as in both techniques, the current is measured by varying the potential applied to the working electrode. In this case, both sensors rely on faradaic processes occurring at the surface of an electrode [39,47]. The main difference between these two types of biosensors is that they are conducted by different interfacial methods.

Amperometric measurements adopt the potentiostat methodology and rely on the principle that under constant electrode potential, the resulting diffused-current, is proportional to the analyte concentration. However, voltammetric sensors adopt a potentiodynamic approach of amperometry, with a potential range swept during the experimental measurement assays [35]. By integrating obtained electric current over time, it provides the charge associated with the redox reaction [48], which may afterwards be related to the amount of analyte that reacted, by Faraday’s 1st law of electrolysis states, described by Equation (2),
(2)$$Q=\overset{t_{f}}{\underset{t_{i}}{\int }}I\;\mathrm{dt}$$
where, Q is the total electric charge (C) passed through the analyte, and I is the electric current measured (A) over time t (s).

Alternatively, the difference of the mass of analyte m [49], resulting from the chemical reaction occurred at the electrode, may be determined according to Equation (3) that describes the Faradaic first law of electrolysis state,
(3)$$m=\left(\frac{Q}{F}\right)\left(\frac{M}{z}\right)$$
where, m corresponds to the altered mass of the analyte (g), at the working electrode, *F* is the Faradaic constant, *M* is the analyte molar mass (g.mol^−1^), and *z* refers to the valence number of the ionic species among the sample of analyte (corresponds to the number of electrons transferred per ion [47].

Depending on the target biomolecules specificities to be detected, there are several of amperometric measurement sensors types [50]. Therefore, the detection of organic molecules involved in metabolic pathways, are attained with the assistance of catalytic enzymes. Such is the case of the first biosensing device; the glucose sensor that relies on Clark’s oxygen electrode [15] for the measurement of free glucose levels in body fluids, first proposed in 1962. This first used sensor was an enzyme-based amperometric device that employed the glucose oxidase (GOx) as enzymatic mediator immobilized at the electrode surface.
(R1)$$C_{6}H_{12}O_{6}{\;+\;O}_{2}\;\overset{\;\mathrm{Glucose}\;\mathrm{Oxidase}\;}{\to }{\;C}_{6}H_{10}O_{6}{\;+\;H}_{2}O_{2}$$

In this amperometric biosensor, the product of the enzymatic redox reaction is diffused to the transducer surface where generates an electrical response. The concentration of this product is attained since it is proportional to the measured electric current [51].

However, firstly used amperometric biosensor in glucose detection had some significant drawbacks:

When the glucose level is sensed through the measurement of the hydrogen peroxide (H_2_O_2_) generated from the biochemical oxidation of β-D-glucose (R1), it requires a high operating potential (0.6 V vs. SCE) to accomplish a high selectivity in detection [52], which results in inaccurate measurements of glucose concentration. Moreover, the oxidative reaction of *β*-D-glucose supervised during the first generation of amperometric enzyme-based glucose detection, involves the presence of the molecular oxygen as an electron acceptor. However, the solubility of such oxygen is quite limited in biological fluids, therefore measurements are thus dependent on atmospheric oxygen that is also subject to natural fluctuations. This inconstant state is defined as oxygen “deficit”, and impacts sensor response, by fostering measurement errors, such as lack of linearity in the return measurement, thereby reducing the linear range of the biosensor [43].

Consequently, different methodologies for signal transportation into transducer surface had to be developed and following the signal mode or electron transfer methods used for biochemical reaction measurements. Therefore, amperometric enzyme-based biosensors have been developed over the so-called three “generations” (Figure 4) [53].

1st generation biosensors: Electrons resulting from redox reaction are transferred to molecular oxygen, and the transduction of bio-recognized redox reaction products or reactants, is conducted at the surface area of the working electrode. Hereby it is measured the decrease in the O_2_ concentration and/or the produced H_2_O_2_.

2nd generation biosensors: Electrons resulting from redox reactions are transferred to a molecule that behaves as an artificial mediator and transports the electrons from the enzyme active site to the working electrode surface, where the transduction of bio recognized reaction takes place. 

3rd generation biosensors: Electrons resulting from redox reactions are directly transferred from the redox-active site of enzyme cofactor to previously modified electrode surface, without any intermediate stages or mediators, as the enzyme is immobilized and in “direct electrical communication” mode with working electrode surface (the transducer).

As a result of the electrode surface modification, the enzyme can communicate directly with the working electrode, and exchange electrons directly with the metal surface of the electrode. 

Amperometric biosensors are electrochemical sensors suitable for rapid and real-time diagnosis of pathogens and are widely used in the prognostic of diabetes, respiratory-related diseases, neurodegenerative or infectious diseases.

The amperometric techniques have widely applied in the detection of glucose [46], H_2_O_2_ [55], L-glutamate [56], bacteriophages or phages [57], Infectious human pathogens - *Streptococcus pneumoniae* [58], *Escherichia coli* [59,60] enterovirus 71 [61], pesticides carbamates [62], eosinophil cationic proteins [63]. 

An electrochemical reaction is a multi-step process that involves the diffusion, ionic migration, and transfer of electrons of the analyte sample, towards/from, the electrode/solution interface. 

Some other biological analysis may be developed utilizing the measurement of the electrical resistance of the organic compounds, through electrochemical impedance spectroscopy.

The quantitative biological detection conducted by the EIS technique is based on the measurement of the electrical properties of the working electrode, after the disturbance of its electrochemical equilibrium state [64], achieved by the application of a minor sinusoidal wave of potential variation through time. 

This detection method provides information on each step involved in the electrochemical reaction of the analysed sample and allows to map charge distribution at the interface between the electrode/solution of the analyte, as well as to establish the electrical resistance of the solution to be analysed. Through the analysis of obtained data is possible to ascertain the rate constant of the electrochemical reactions involved, as well as the diffusion processes associated with the transport of the analyte to the surface where this reaction occurred. 

The most relevant parameters obtained by impedance spectroscopy [36]:The ohmic resistance of the analysed solution,Capacitance and charge distribution, of the interface electrode/solution,The electrochemical reaction rate-constant.

Consequently, obtained impedance values stem from the conducting state of the electrode material, wherein modified electrodes are frequently used in the electrochemical measurements, with several components amending electrode catalytic performance. It is essential to distinguish the influence of those components in the electrochemical reactions of the analysed sample. Each type of coating material used on the electrode surface (carbon support, polymer, nanomaterials, enzymes, etc.) has an impact on the impedance plots and reflects the current flow capacitances and impedances [53].

Therefore, electrochemical impedance spectroscopy is a valuable method to follow the recognition processes used in biological detection.

Impedance-based biosensors can be established by integrating a sensing element into an electrochemical cell. This is managed by using a three-electrode cell set-up, where an electric small sinusoidal disturbance/perturbation (potential or current), is applied to the system to alter its state of equilibrium. As the electrical pulse applied is extremely small, to avoid any external interference np the impedimetric measurement process, a protective shield is used, namely the Faraday cage (which protects electrochemical set up from electromagnetic fields).

Tae-Hyung Kim et al. [65] developed a highly sensitive dopamine detection method that can be highly useful for early diagnosis of various neurological disorders, like Parkinson’s, and schizophrenia, attention deficit hyperactivity disorder (ADHD) and drug addiction [66]. They reported a cylindrical gold nanoelectrode (CAuNE) platform as an amperometric dopamine biosensor, with a linear range of 1–100 μM of dopamine, in the presence of glucose (40 g L^−1^), uric acid (44 mM) and human serum albumin (0.1 g L^−1^), with an LOD of 5.83 µM, suitable for the sensitive detection of neurotransmitters released from human neuronal cells.

Impedance techniques are swift and convenient approaches to explore the electrical behaviour of diverse electroactive materials, thereby relevant in monitoring certain pathogens as well as electrochemical systems used in their detection [51,52] such as:Impedimetric detection of biological molecules and inorganic materials, under the employment of alternating current. Herein, the identification and quantification of biological molecules and inorganic species are accomplished by using the analysis of electrical impedance variation, arising from the small electrical disturbance applied to the electrochemical system under equilibrium. Growth of the impedance curve allows a rapid counting for the detected species, as the charge transfer resistance (Rct) is proportional to analyte species concentration. Impedimetric detection has been widely used in bacteria detection, heavy metal contamination counting, and aptamer [67].Electrochemical system characterization/assessment to the catalytic material employed in the electrode/for exploring the change in electrical behaviour of diverse materials used in modified electrodes. Throughout EIS it is also possible to determine the ionic diffusion (ionic transport) of various ions within the material structure of the electrode [68]. Therefore, EIS can be either used as a diagnostic tool to find an optimised nanocomposite loading, for a given new catalyst to be used on several biomolecules reduction/oxidation reactions, with the aim of their detection.

Recently, Donghai Lin et al. [69], have developed a simple and highly sensitive impedimetric immunosensor, based on the affinity reaction antibody-antigen establish on a paper electrode surface, for the sensitive detection of *Escherichia coli* O157:H7 bacteria. The integration of structural and electrical properties of reduced graphene oxide paper with a large active surface area of gold (Au) nanoparticles, for antibody immobilization, has shown a highly specific and stable detection of *E. coli* O157:H7, with a wide linear detection range 1.5 × 10^2^ to 1.5 × 10^7^ colony-forming unit.mL^−1^, with a correlation coefficient of 0.9805.

## 7. Enzyme-Based Recognition Elements for Electrochemical Sensors

The main factor underlying the sensitivity and selectivity of an electrochemical biosensor concerns the performance of the transducers used. Biosensor response relies on the bioreceptor’s surface ability to adsorb the biological molecules of interest that promptly react at the active site, to transfer associated electrons, and thus establish the electronic signal launched by the transducer [70].

Based on the resulting initial reaction rates, the enzymes involved in the physiological processes of the human body are commonly used in most of the electrochemical detection transducers, as in the determination of the concentration of their substrates. Such is the case of the potentiometric glucose sensing, with the entrapment immobilization of the glucose oxidase (GOx) into polymeric films [71].

Generally, enzymes are the most common and developed used bio-recognition receptors in the surveillance of biological species [54]. Along with antibodies, enzymes are naturally occurring bio-recognition elements, in living organisms. Usually, the enzyme is a protein that functions as a biological catalyst―e.g., substance that speeds up a chemical reaction without being changed or depleted by the reaction. As biologically derived, enzymes can take advantage of their naturally adapted physiological interactions to most efficiently achieve analysed substrate’ specificity, aiming the catalysis of their redox reactions.

The configuration of a biosensor transducer involves the suitable immobilization of the enzyme on its surface. This required immobilization process [54] is a critical step in the design of biosensors and attained through different bonding forms.

Thus, enzymes are immobilized onto an electrode surface, by adsorption, affinity, covalent binding, cross-linking, by electrostatic attraction, or even entrapped into a polymeric net structure or polymerized gel, as well as the combination of them. This mobilization procedure has the goal to ensure the proximity between the active site of the enzyme and the conducting surface of the electrode, thereby improving the performance of biosensors. The enzymatic active site consists of a protein section where the substrate temporary binds (binding site) and the catalytic reaction of the substrate occurs [72].

As proteins of globular structure, enzymes are impaired by fluctuations in temperature and pH, which adversely affect their tridimensional conformations and consequently its active sites catalytic activity [73].

For the enzymes immobilized at the electrode, such detriments result mainly from protein denaturation and consequent deactivation, which ultimately reduces the useful life of the sensor when adsorbed at the electrode surfaces [53]. The adsorption of denatured proteins leads to the bioactivity loss.

Also during the immobilization process itself, there is a high probability that enzyme denatures, due random distribution and/or incorrect dispersal of proteins onto electrode’ surface, and causing a misalignment of the proteins’ redox centre concerning the electrode surface, which may block entirely or partially the active site of the enzyme [53].

In general, there are two main failures in enzymatic/mediated biosensing, concerning the decline in its catalytic activity [53]:When the conductive material detaches electrode surfaceWhen the electrode experiences biofouling or passivation.

The enzymatic impairment most frequently found in biosensors and the most common cause of biosensor failure in vivo applications is the biofouling [53]. By experiencing biofouling, proteins, cellular debris, or living cells adhere or adsorb to biosensor outer surface, this way impeding analyte diffusion at the biosensor surface, which ultimately leads to a decrease in sensor response. Moreover, there is passivation of the electrode, when small molecules are transformed into adherent substances at the electrode surface, reducing its active area [53].

Additionally, the lifetime of enzyme-based biosensors is closely related to the loss of enzyme activity over time [53]. There are therefore two fundamental requirements in the development of a biosensor, its: 1ifespan, and operational stability.

Storage conditions during the time lag between its manufacture and its operation can affect the enzymatic activity of the biosensor.

Operational stability concerns the ability of biosensor’ immobilized enzymes to retain activity during their operation; this capacity has an impact on the operability and reproducibility of the biosensor.

Although most studies and applications of electrochemical detection of biological species involve the presence of catalytic enzymes. These biosensors are often affected by the above-mentioned stability problems, which are intrinsically related to the enzyme nature [74]. To surmount this limitation, electrochemical detection studies have been performed on electrodes without enzymes immobilized on their surface [75].

Consequently, the most promising approach for the development of electrochemical biosensors is the establishment of a direct electrochemical interface between the biomolecules and the electrode surface. Bearing in mind that selectivity is a crucial issue in non-enzymatic biosensors, it is this way essential to identify a suitable electrochemical conductive material to trace some of the biological species of concern selectively.

Accordingly, polymeric films, metallic oxides as well as metallic alloys have been employed in electrochemical biodetection in clinical assays [76], achieving high precision selective results from biological samples of medical relevance.

Given the feasibility of non-enzymatic detection with high screening performance, there is only necessary for the development of bio-recognition conducting elements, able to improve sensitivity along with selectivity in molecular detection [77].

The recognition element of an electrochemical probe corresponds to the catalytic material use at the surface of the working electrode employed to quantify the electroactive species of the biological sample. In a non-enzymatic assay, this material is composed of a layer of conductive material in the proximal vicinity of the electrode [47].

Among the most conductive material employed in electrochemical probes for catalytic applications are nanoscale metallic compounds. Such as metal/metallic-alloys nanoparticles.

## 8. Overview of Nanoparticle as Electrocatalysts

The key factor underlying an electrochemical biosensor sensitivity and selectivity is the electron driving performance of the materials employed at electrode (probe) surfaces.

In general, high electrical conductivity materials such as metals and metallic oxides might easily promote the electron transfer on electrochemical devices. Especially noble metals have proven to be highly conductive catalytic materials for the improvement of electrochemical performances, with platinum (Pt) as one of the most widely used high-cost noble metals [78,79].

In addition, electrochemical catalytic activity is sensitively dependent on the dimensions of the catalyst. Therefore, the biosensor sensitivity underlying electrode catalytic activity improves as the catalyst surface area increases.

Nanostructured catalytic materials employed in sensor electrode’ surfaces have demonstrated promise in increasing the sensitivity of electrochemical biosensors due to the:Increased surface area to volume ratio,Increased density of bio-recognition elements (active sites),Improved access of target molecules to bio-recognition elementsHigher catalytic activity

The underlying reason for changes in the properties of solid materials, along with their size is the ratio high fraction of atoms on materials’ surface to the total amount of atoms in the materials’ mass. In a smaller size particle, this ratio increases, turning more significant the atoms at the surface. 

Accordingly, nanomaterial’s electronic energy states start to behave in discrete mode, with thermodynamic laws no longer meaningful and consequently leading to the observed unique physicochemical properties in nanomaterials. Specially diverging in electronic, optical, magnetic, and mechanical characteristics.

Conventional thermodynamics laws applied for bulk materials no longer suits nanometre-scale materials. For instance, properties like entropy, enthalpy, free energy, melting temperature, ordering temperature, Debye temperature, and specific heat, no longer remain constant but vary according to solid crystal dimension size, and morphology [80].

As the material size approaches the electron mean free-path, the electron transfer between the atoms comprising the material becomes more unlikely, as the electron transfer d-bands of the several atoms are further apart [81].

Usually, bulk scaled transition metals, exhibit a high electrical conductivity due to their large d-band, yet such characteristic electrical conductivity fade as the particle size diminishes.

However, while individual metal nanoparticles exhibit lower electrical conductivity, their high surface area to volume ratio compared to the corresponding bulk-state, allows such metallic nanoparticles exhibit distinctive skills in the area of catalysis [82]. This one-of-a-kind property is responsible for the increased number of active sites for electron transfer to occur, making the catalytic approach more efficient [83].

An extensive range of solid nanomaterials has been widely used as catalysts in a high sensitivity detection of biological species. Namely transition metals [84], metal oxides [85,86], metal alloys [87], carbonaceous materials [88,89], zeolites [90] and polymers [91].

Particularly, in heterogeneous catalysis, platinum nanomaterials have been quite analysed as promising material, since its’ porosity supplementary feature, enhances their surface area and consequently the electron transport capacity. This makes platinum nanomaterials much amenable to H_2_ adsorption [92].

Due to their relative-inert nature, noble metal nanoparticles are preferred over other transition compounds, in the construction of electrocatalytic materials, since their filled d-subshell electrons are more vulnerable, promoting the dissociative adsorption of species at their surface.

Therefore, nano-sized materials, composed of various noble metals have shown high catalytic performance when used as a bio-recognition conductive material (i.e., exhibit better catalytic activity).

In particular, it was attained a low detection limit (3 µM), for the amperometric detection of H_2_O_2_, at the surface of dendritic nanoporous gold [93] material, using a low electrode potential value of −0.1 V vs. Ag/AgCl.

Also, other literature results have been reported to employ other metallic nanostructured materials such as silver, palladium, ruthenium, rhodium, iridium, and osmium, leading to the attainment of quite low detection limits [94,95,96,97].

However, there are also a quite number of reactive transition metal ions with variable oxidation states, which can both give and accept electrons easily, thereby making them very good catalysts. The reason relies on the unpaired-valence electrons instability that fosters molecular reactivity and consequently reduces the required activation energy for the reactions to occur [62]. For instance, bimetallic nickel-cobalt sulphides (${\mathrm{NiCo}}_{2}S_{4}$) within reduced graphene oxide rGO net, exhibit a low detection limit of 0.19 µM to the reductive H_2_O_2_ detection [98]. Besides, polyoxometalates high-valence transition metal oxy-anions, linked by oxygen atoms and forming a massive 3-D framework, reveal to be very sensitive to the detection of ${\mathrm{NO}}_{2}^{-}$, ascorbic acid (AA), and dopamine (DA), with the lowest detection limits of 0.45 µM, 0.03 µM, and 0.18 µM, respectively [99].

## 9. Reactive Species under Physiological Metabolism

In a general sense, oxygen molecules (O_2_) are crucial for eukaryotic organisms, with atmospheric oxygen accounting for energy generation from the respiration of all aerobic living organisms. 

In its triplet ground state, the molecular oxygen remains non-reactive, and the ability of this molecule as an oxidant is somewhat restricted, as it is only feasible when unpaired electrons are antiparallel to each other [100]. This is an unlikely event, since in its ground state, diatomic molecular oxygen, has already two unpaired, parallel-spin electrons in two different molecular valence orbitals.

Such a spin constraint means that, on its own, molecular oxygen is not sufficiently reactive to capture electrons from other biochemical species. However, this spin restriction vanishes when molecular oxygen is reduced, in a single electron transfer, forming the initial reactive oxygen species (ROS), the superoxide anion, such as illustrated by the electron transfer chain process in Figure 5.

Due to its unique biradical character, each monoatomic oxygen atom that composes O_2_ molecule exhibits one unpaired electron, as observed in different types of reactive structures and intermediates (such as superoxide), which are referred to as reactive oxygen species (ROS). In such a state, ROS become transient, unbalanced and extremely reactive, able to combine with other atoms also containing unpaired electrons. Such a condition triggers the formation of further unstable, reactive species in a chain reaction procedure. 

The transition from unreactive to a reactive and transient variant of molecular oxygen occurs in the course of the cellular mitochondrial respiration (MR) process, wherein the molecular oxygen is a receptor of the electrons flowing across the electron transport chain process (ETC) aiming the creation of a charge gradient between both sides of the inner mitochondrial membrane.

The essential function of mitochondria is to accomplish cellular respiration straight after the nutrient breakdown during glycolysis and the citric cycle and turns them into energy. This energy is produced in the form of adenosine triphosphate ATP nucleotide and will be then used by the cell to carry out various body’s cellular functions.

This mitochondrial metabolic process starts at the citric acid cycle, tricarboxylic acid cycle (TCA), or Krebs’ cycle. This cycle is made of a series of chemical reactions with the main goal of generating energy in ATP form. Above all, it firstly oxidizes pyruvate to form acetyl-CoA, and over the cycle will several times reduce NAD^+^ to NADH and produce carbon dioxide (CO_2_). Such acetyl-CoA is a metabolite derived from carbohydrates, lipids, and proteins catabolism [101], which is converted into succinate at the fifth step of Krebs’ cycle, producing one ATP molecule by the phosphorylation of adenosine diphosphate (ADP) nucleotide.

This cyclic process depends on the mechanical rotary action of complex F0F1-ATP synthase that in turn, is stimulated by the potential difference at the inner mitochondrial membrane, which occurs during oxidative phosphorylation (OP). Potential gradient development results from the electron transfer that takes place along a chain of four protein complexes locally embedded into phospholipid inner mitochondrial membrane referred to as ETC.

The purpose of this electron transport chain is to induce the creation of a charge concentration gradient between both sides of the inner mitochondrial membrane. Given its electronegativity, molecular oxygen is thus amenable to capture electrons and, therefore, receives the electron leftovers at the end of the ETC, thus becoming a transient and reactive species.

Although one of the main goals is to reduce molecular oxygen directly into two molecules of water. Inevitably, along the course between the four protein complexes, there may be as well an incomplete reduction of the oxygen molecule, leading to the formation of oxygen species electronically unbalanced, therefore extremely reactive.

The primary reason for eukaryotic species requires molecular oxygen, which is their constant usage as a necessary electron acceptor at the end of ETC during mitochondrial respiratory metabolism. Being this a key factor for the creation of a potential difference between the inner mitochondrial membrane matrix side and the intermembrane space [102]. The resulting gradient will boost F0F1-ATP synthase mechanism, embedded within the mitochondrial inner membrane, for the release of ATP energy, the universal energy donor in the cell, fundamental to all the functions of the cells of living organisms. Due to this dependence on molecular oxygen, mitochondrial metabolism is also referred to cellular mitochondrial aerobic respiration (MAR).

Accordingly, the reactive oxygen species (ROS) formed during MAR, are biological molecules derived from the cellular metabolism resulting from a series of processes that occur in the cells and tissues of the human body.

Specifically, such biochemical set of reactions that generate ROS occurs on several organelles of eukaryotic cells, such as mitochondria, peroxisomes, and endoplasmic reticulum, or even the enzymatic action of nicotinamide adenine dinucleotide phosphate (NADPH) oxidase [103,104].

The ETC process, portrayed in Figure 5, aims the creation of a positive charged transmembrane concentration gradient of protons that will spin F0F1-ATP synthase complex, to spark catalysis of ADP phosphorylation, into ATP. At the end of the ETC process, transferred electrons reach complex IV, and for every four electrons transferred, four protons provided by the mitochondrial matrix region are joined to react with an oxygen molecule and yield two water molecules.
(R2)$$O_{2}{\;+\;4e}^{-}{\;+\;4H}^{+}\;\to {\;2H}_{2}O$$

Unbalanced charge of generated ROS stems from the leakage of electrons in any of the four redox-centres that form ETC. Resulting in an insufficient reduction of molecular oxygen, corresponding to less than four electrons for each oxygen molecule, and consequent formation of oxygen-based reactive species in the mitochondrial matrix (as represented in Figure 6). As a consequence of this electron spill, there is a decrease in the efficiency of the oxidative phosphorylation (OP) process, which increases as a result of aging or pathological conditions. 

Due to the electrons leaked during the ETC of OP, different ROS will outcome from the sequence one-electron transference to molecular oxygen, yielding the prevalent endogenous ROS (shown in Table 2).

During the ETC process, the electrons are forwarded along through different protein complex clusters, owing to the electrochemical motive force, generated during the oxidation-reduction reactions at each complex, which exhibits a more negative reduction potential than that of the preceding complex.

Effectively, over the ETC course, each protein-complex receptor, exhibit a higher reduction potential than what was achieved in the preceding complex.

Under physiological conditions, during the ETC journey, 0.2–2% of the electrons do not follow the regular transfer order, but leak out and directly interact with molecular oxygen in some incomplete reduction’ side-reactions, to produce superoxide or any other ROS derived intermediates (Figure 7).

Protein clusters involved in the mitochondrial ETC, have probable sites for ROS generation, and there are up to 11 distinct sites where ROS can be eventually produced in isolated mitochondria [105]. Some of them located within the inner mitochondrial membrane and others in the mitochondrial matrix.

Throughout four protein-cluster ETC complexes, there are specific sites of production of superoxide and hydrogen peroxide (molecular oxygen one-electron reduction and two-electron reduction, respectively), directly involved in the oxidative degradation of the substrate (also mentioned mitochondrial ROS or mROS).

When electron leakage takes place during ETC there are several sites where ROS formation is favoured, called the sites of ROS production [97].

In complex I there are two-site of radical production; sites *I_F_* and *I_Q_*. Sites *I_F_* is thought to predominantly generate oxygen one-electron reduction, superoxide anion radical, whereas in site *I_Q_* the electron leak may produce a mixture of superoxide radical and hydrogen peroxide.

In complex II, there is one site *II_F_* that may generate both superoxide and hydrogen peroxide. 

Under normal conditions, the amount of ROS produced in site *II_F_* is negligible, nevertheless, there has been observed an increase in the ROS derived from the *II_F_* site, in individuals with Complex II mutation-related diseases.

In complex III, there is one site *III_QO_* that is thought to predominantly produce superoxide anion.

Typically, in complex IV the molecular oxygen is either attached to a heme-a3 metalloprotein or negatively polarized, therefore unavailable for uncompleted electron reduction. As a result, complex IV is less prone to produce ROS, and four electrons along with four protons bind the dioxygen molecule into two molecules of water (R2).

The leakage of electrons during ETC, and consequent early incomplete reduction of molecular oxygen, is the reason behind ROS generation during OP (Figure 7).

## 10. ROS Sources

Following their origin, ROS may be either endogenous when generated intracellularly or exogenously when granted by external causes:

### 10.1. Endogenous ROS

Endogenous ROS are ubiquitous species generally developed during the ETC course of OP process that play antagonistic roles on the human organism, depending on the levels found in the body fluids.

Under physiological conditions, generated ROS are predominantly beneficial for cells and have some purposeful roles in the biology of living organisms with essential tasks in support of several physiological operations/functions. Therefore, ROS is essential for several regulation processes occurring in the human organism, like cell-growth regulation, and intercellular signalling, as well [106].

Fundamentally, the ROS family formed in eukaryotic organisms’ have the roles:

1. ***Regulation***: 

To regulate cellular proliferation, differentiation, and apoptosis (programmed cell death).

2. ***Protection***:

To protect the human system from cytotoxic species, as well as counter their infections, by reacting with the adverse microorganisms to breakdown them and neutralize their damaging effects.

3. ***Management***:

Rules some functions of circadian rhythm physiology, like body temperature, sleep-wake cycles, metabolism, blood pressure, and guides the cycles of several hormones’ secretion.

#### Nitrogen Species

Likewise, oxygen molecules, also nitrogen is essential to eukaryotic organisms as it makes part of the amino acids that frames proteins. This way it is also needed to assemble nucleic acids that compose DNA and RNA.

Among higher vertebrates, nitrogen is present in the form of nitrogen monoxide, or nitric oxide molecule (${\mathrm{NO}}^{●}$), and is known to play an important role in many physiological processes.

Nitric oxide is an intra- and extracellular messenger molecule that mediates diverse signalling pathways in target cells, and is known to play an important role in many physiological processes including, neuronal signalling, immune response, inflammatory response, modulation of ion channels, phagocytic defence mechanism, and cardiovascular homeostasis.

Therefore, it is intimately involved in regulating many physiological processes of humans’ life as walking, digestion, sexual function, perception of pain and pleasure, memory recall, and sleeping [107]. Whenever reactive transient species are capable of independent existence, they are pointed as free radicals. Nevertheless, not all ROS are free radicals, and some reactive species may not have any unpaired electrons among their valence shell, as hydrogen peroxide (H_2_O_2_) and peroxynitrite (ONOO^−^) [108].

ONOO^−^ belongs to the category of reactive species of nitrogen (RNS) and is a very potent oxidizer that plays various pathophysiological roles in the development of inflammation [109]. Several lines of evidence support that ONOO^−^ is a potent cytotoxic involved in the pathogenesis of several chronic diseases such as diabetes *mellitus* and CV dysfunction [103].

In the same way as ROS, also RNS may have a detrimental effect when under stressful amounts within body fluids, with NO having a paradoxical role in their properties. Therefore, under healthy physiological conditions NO has a beneficial role, acting as a regulatory agent of the CV system, or else, under pathological inflammatory conditions, owns a detrimental assignment in triggering chronic inflammatory pathway. In such deleterious conditions, ${\mathrm{NO}}^{●}$ reacts with${\;O}_{2}^{●-}$, and originates the harmful reactive ONOO^−^ [110].

Therefore, ROS and RNS, that participate in normal cellular function or pathological mechanisms (depending on their overproduction), are produced through several mechanisms by the cell, during the electron transport chain in mitochondria, through various cytosolic and membrane enzymes (i.e., xanthine oxidase (XO), nitric oxide synthase (NOS), NADPH oxidase complex, etc.).

### 10.2. Exogenous ROS

Organism’s external factors, such as metabolites and chemicals that emerge from exposure to environmental pollutants, radiation, cigarette smoking, certain foods, and drugs, considerably stimulates the formation of ROS. These reactive forms are made in response to ultraviolet (UV) radiation, cigarette smoking, alcohol consumption, ingestion of nonsteroidal anti-inflammatory drugs (NSAIDs), and many other exogenous agents.

Effectively after the incursion into the body, certain environmental pollutants, such as carbon monoxide, nitric oxides, and transition metals, act as powerful oxidizers, by stimulating a wider generation of ROS in biological systems, thereby promoting cellular oxidative stress [111].

Whenever tobacco smoke is breathed in, numerous extremely reactive chemicals enter the bloodstream, producing extensive oxidative damage to cellular protein structures, resulting in high carcinogenic and toxic potentials for the living organism [112].

High-level blood alcohol, alter the number of certain metals in the circulatory system, that stimulates the catalytic activity of the enzymes that assist the formation of ROS, and at the same time, reduces the amount of some agents involved in the neutralization of ROS species [113], fostering the development of ROS.

Moreover, the ingestion of certain foods [114] and painkiller drugs [115] promotes an increase in the enzymes involved in ROS formation that participate in the attack against microorganism/toxin assault and mitigates mitochondrial respiratory function by disturbing the transport of electrons during ETC, thereby lowering the production of ATP, and increasing ROS generation, respectively.

## 11. The Exponential Growth of Highly Reactive Species among the Human Body

Under physiological conditions, ROS is mostly developed in the course of the ETC mitochondrial process, which is initiating in the complex I when electrons assigned to be transferred to the metal cofactors Fe-S cluster, leak, and partially reduce some oxygen molecules into superoxides. Such a process is assisted by the enzymatic action of NADPH oxidase, which can transport the electrons across the plasma membrane and generate superoxide and other downstream reactive oxygen species [116].

Besides, ROS may be either formed by other cellular organelles, such as cytochrome P-450 and peroxisomes. These are membrane-based cellular organelles of high oxidative capacity to produce mainly H_2_O_2_.

Accordingly, H_2_O_2_ is generated [117] during the cytochrome P-450 monooxygenase cycle of alcohol metabolism, which occurs in the liver under the assessment of enzyme cytochrome P450, or even by peroxisome membrane-organelles [118] present in the cytoplasm of eukaryotic cells, which are involved in the decomposition of very-long-chain fatty acids, branched-chain fatty acids, and amino acids [119].

On the other hand, ROS generation may even be induced by the stress condition of some membrane-organelles; such is the transportation system of the cell the endoplasmic reticulum (ER). During a stressful situation, the network of tubules that constitutes endoplasmic reticulum membrane-organelle, responsible for protein biosynthesis and folding, is also responsible for the increment of ROS species.

Under such a stressful situation, endoplasmic reticulum tends to boost the regular release of calcium ions Ca^2+^ into the cytosol, this way enhancing the concentration of these ions within the mitochondria. Given that, mitochondrial Ca^2+^ activates key enzymes involved in the ATP synthesis, the generation of mitochondrial reactive species during the ER stress state increases [120].

The produced, short-lived intermediate oxygen reactive species ($O_{2}^{●-}$, H_2_O_2_ and OH●), participate in redox reactions that lead to an oxidative change in several biomolecules, such as proteins or lipids as prime targets.

Under physiological conditions, the early formed $O_{2}^{●-}$, present in cellular aqueous media: 

Suffers a spontaneous dismutation into hydrogen peroxide (H_2_O_2_), over the catalytic effect of manganese superoxide dismutase (MnSOD) metalloprotein [121].
(R3)$$O_{2}^{●-}{+\;2H}^{+}\overset{\;\mathrm{MnSOD}\;}{\to }{\;H}_{2}O_{2}$$

1Reacts with available free radical nitric oxide (NO^●^) in a diffusion-limited reaction catalysed by endothelial nitric oxide synthases (NOS), to yield another powerful biological oxidant: the peroxynitrite anion (ONOO^−^) [122]:
(R4)$${\mathrm{NO}}^{●}+O_{2}^{●-}\to {\;\mathrm{ONOO}}^{-}$$
2Reacts with existing hypochlorous acid (HOCl), and generates hydroxyl radical (OH), according to the reaction mechanism [123]:
(R5)O2●−+HOCl → O2+Cl−+OH●
3H_2_O_2_ resulting from the partial reduction of O_2_:
(R6)H2O2+O2●−→OH●+OH−+O2


Is decomposed into hydroxide anion (${\mathrm{OH}}^{-}$) and into the most reactive compound, the hydroxyl radical (O●H). Such reaction, which is generally very slow in an aqueous environment, when in the presence of free ionic metals such as ferrous ${(\mathrm{Fe}}^{2+})$ and cuprous ${(\mathrm{Cu}}^{+})$, proceeds very quickly, and is designated by metal-catalysed Haber-Weiss reaction [124].

Effectively, both iron (Fe) and copper (Cu), are essential nutrients in a human organism with very common existence in low-affinity complexes of sulphur clusters (Ion-S) of intracellular surroundings, thereby considered redox-active labile ions [125]. Both ions are oxidized in the presence of H_2_O_2_, however, reduced by $O_{2}^{●-}$ in a Fenton reaction mechanism. When Fenton redox reaction use the couple ferrous/ferric ions ${(\mathrm{Fe}}^{2+}/{\mathrm{Fe}}^{3+}):$(R7)$${\mathrm{Fe}}^{3+}{\;+\;O}_{2}^{●-}\;\to {\;\mathrm{Fe}}^{2+}{\;+\;O}_{2}$$
(R8)Fe2+ + H2O2 → Fe3+ + OH● + OH−

Then in the presence of ${\mathrm{Fe}}^{3+}{/\mathrm{Fe}}^{2+}$, OH● is further generated through Haber-Weiss reaction:(R9)O2●− + H2O2 → O2 + OH● + OH−

Thereby, the result endogenous ROS exponential growing in consequence of the impartial reduction of oxygen along with the autoxidation of primarily produced $O_{2}^{●-}$, and H_2_O_2_.

4Reacts with chloride ion (Cl−) when driven by the respiratory burst of the immune system, in a reaction catalysed by myeloperoxidase heme-enzyme (MPO), producing the reactive compound hypochlorous acid (HOCl) to be used during the phagocytosis process [126].
(R10)Cl−+H2O2→  MPO   HOCl+O−H 
5Undergoes catalytic reduction into water and oxygen molecules through several reactive oxygen species scavengers, responsible for its deletion. Thereby cellular level of H_2_O_2_ are sequentially controlled by some oxidoreductases (enzymes that catalyses both the oxidation and reduction reactions):
aCatalase (**CAT**), intracellular, heme-based soluble enzyme - locally confined to peroxisomes and conversely to the other two GPxs and Prxs, does not require any reductase to endure H_2_O_2_ decomposition process (both GPx and Prx, rely on reductases* as electron donors) [127]:
(R11)$$H_{2}O_{2}\;\;\;\overset{k_{\mathrm{CAT}\;}}{\to }\;\;\;H_{2}O+\frac{1}{2}O_{\begin{array}{c} 2 \end{array}}$$* Glutathione reductase (GSH) or Thioredoxin (Trx)bGlutathione peroxidase (**GPx1**), a cytosolic antioxidant enzyme that reduces H_2_O_2_, and other organic peroxides, into the water and to water and lipid alcohols, respectively, through [124]:
(R12)$$H_{2}O_{2}+2\mathrm{GSH}\;\overset{\;\mathrm{GPx}\;\;\;\;\;}{\to }{2H}_{2}O+\mathrm{GSSG}$$
(R13)$$R-\mathrm{OOH}+2\mathrm{GSH}\;\overset{\;\mathrm{GPx}\;}{\to }{\;\mathrm{ROH}+H}_{2}O+\mathrm{GSSG}$$
(R14)$$2\mathrm{GSH}\;\overset{\;\;\;GR\;\;\;}{\leftarrow }\mathrm{GSSG}$$
(GPx–Glutathione peroxidase; GSH–Glutathione, GSSG–Glutathione disulphide (two molecules of the oxidised form of GSH, GR–Glutathione reductase)cPeroxiredoxins (2-Cys **Prx**), a cytosolic enzyme that eradicates H_2_O_2_ through its oxidation of catalytic cysteine residue at position 47 on Prx, and responsible for the peroxidase activity [124]. Both typical and atypical 2-cys types:
(R15)$$\mathrm{Prx}\left(\mathrm{SH}\right)\left(\mathrm{SH}\right){+H}_{2}O_{2}\overset{{\;k}_{p1\;}}{\to }\mathrm{Prx}\left(\mathrm{SOH}\right)\left(\mathrm{SH}\right){+H}_{2}O$$
(R16)$$\mathrm{Prx}\left(\mathrm{SOH}\right)\left(\mathrm{SH}\right)\overset{{\;k}_{p2\;}}{\to }\mathrm{Prx}\left(S\right)\left(S\right){+H}_{2}O$$
(R17)$$\mathrm{Prx}\left(S\right)\left(S\right)+\mathrm{Trx}\left(\mathrm{SH}\right)\left(\mathrm{SH}\right)\overset{{\;k}_{p3\;}}{\to }\;\mathrm{Trx}\left(S\right)\left(S\right)+\mathrm{Prx}\left(\mathrm{SH}\right)\left(\mathrm{SH}\right)$$
(R18)$$\mathrm{Prx}\left(\mathrm{SOH}\right)\left(\mathrm{SH}\right){+H}_{2}O_{2}\overset{{\;k}_{p4\;}}{\to }\mathrm{Prx}\left({\mathrm{SO}}_{2}H\right)\left(\mathrm{SH}\right)$$


The ‘peroxidation’ cysteine residue of Prx is the site where oxidation by peroxides like H_2_O_2_, lipid peroxide, or peroxynitrite, occurs.

OH● generated by the partial reduction of O_2_:

Hydroxyl radical (OH●) is the most reactive compound and interacts (at the diffusion-limited rate) with almost anything, it contracts with, being able to react effectively with all organic and inorganic matter, cell constituents of eukaryotic living organisms. Namely: proteins: enzymes; lipids, phospholipids; and other metabolites.

In the aqueous phase, OH● has a high affinity for electrons, which predisposes the capture for electrons. In this respect, the majority of such radical reaction with either proteins, lipids or carbohydrates, in high rate constant values (10^6^–10^8^ M^−1^s^−1^), according to three possible reaction modes:1.Hydroxylation: Electrophilic addition of OH● to an unsaturated bond (occurs especially with aromatic-ring compounds);
(R19)ArH + OH●→ ArHOH●2.Hydrogen atom abstraction (typical for alkanes and alcohols);
(R20)RH+OH●→R●+H2O
3.Electron transfer (usually in the presence of inorganic substrates or halides).


Commonly, radical species react extremely fast with other biological compounds, giving rise to the exponential growth of highly reactive unstable species in the human body. To that effect, the oxidative damage follows three steps of action: The initiation; propagation; and termination process. 

Usually, the initiation stage kicks off with the hydrogen atom abstraction, preferentially from the bis-allylic position of the aromatic ring/aliphatic double bond chain. Leading to the formation of a carbon-centred radical R●, or after the release of a molecule of H_2_O, when hydroxylation takes place, as indicated in the RXX.

Under the propagation phase, the alkyl radical reacts extremely rapidly with the molecular oxygen, leading to the formation of peroxyl radicals (ROO•).

For instance, in previous aromatic ring hydroxylation, in the subsequent step, it reacts with a molecule of oxygen, as:(R21)$$A_{r}{\mathrm{HOH}}^{●}{\;+\;O}_{2}\;\to {\;A}_{r}{\mathrm{OH}\;+\;\mathrm{HO}}_{2}^{●}$$

Free radical oxidative chain reaction endures until a less unstable non-radical species is developed. This is accomplished through the assistance of an electron donor compound that features an electron donation capacity to unstable radicals. This kind of support molecules labeled, as antioxidants are electron-charged stable molecules that usually donate a hydrogen atom to the peroxy radical species, resulting in the formation of non-radical products. At this point, it is achieved the termination step of the oxidative damage chain reaction process.

Certain types of organic materials are particularly prone to autoxidation (oxidation that occurs in presence of oxygen), including unsaturated compounds that have allylic or benzylic hydrogen atoms (i.e., located in a site adjacent to an unsaturated carbon atom).

Especially, fatty acids with two or more double bonds readily undergo autoxidation due to the ease by which a hydrogen atom can be abstracted from the unsaturated molecule, and this way initializing the lipid peroxidation process (featured in the Figure 8).

In addition, protein backbone polymeric structure can be attacked on the α-carbon of amino acids, resulting in the formation of corresponding carbon-centred radical R●, that after coupling with OH●. The resulting hydroxylated α-carbon undergoes peptide backbone cleavage at the N–C bond, resulting in a protein conformation change. 

The ROS generated during mitochondrial metabolism are oxidizing agents, which can accept electrons from other biological species, thus disturbing their normal physiological functionality.

As ROS, also NO has a dichotomic role in physiological properties and thus may be either beneficial as a regulatory agent of the CV system or detrimental when under pathological inflammatory conditions, where it reacts with superoxide radical generating a harmful reactive nitrogen species, like the peroxynitrite [128].

RNS act together with ROS in oxidative cellular damage, where it affects all macromolecules (lipid membrane, proteins, and DNA), through lipid peroxidation, by spoiling of structural and enzymatic proteins, and oxidative DNA damage.

## 12. Homeostatic Imbalance of Oxidative Stress

Although ROS can have a destructive effect when are present in higher concentrations, endogenously produced ROS under physiological concentrations can act as intermediates in intercellular signalling, which is extremely important for cellular homeostasis maintenance.

As already stated, whenever present in basal amounts, ROS is essential for different cellular processes, including cell signalling, cell proliferation, and differentiation, adaptation to stress, and metabolic adaptation [129].

Under physiological conditions, the eukaryotic organisms present specific values concerning constituting (metabolites’) substance’ concentrations, and other associated physicochemical parameters, such as body temperature and blood pressure. In a regular physiologic state, there is a permanent and continuous dynamic exchange between the organism and its surroundings’ so that there is an entrance of cellular functions’ required compounds and the release of organic disposable substances. This dynamic mechanism keeps the system balanced and controls the amounts of chemical compounds and related physicochemical parameters that contribute to the regular functionality of the human organism.

The management of an organism’s homeostatic state is accomplished by two factors, which renders its balance: a variable element, and a controlled one. As variable factors, there are those elements that can be adjusted and controlled, such as the biochemical species concentrations and the physicochemical parameters. For instance, the concentration of calcium in the blood or body temperature. The flow rate of biochemical species that transit over the human body is deemed a controlled factor, as it may be handled to manage the variable factors.

Whenever the human organism experiences an inflammatory stimulus, the immune system activates some kind of cells that induce the higher O_2_ uptake, and the associated cellular metabolic activity, during which, the level of certain reactive species (ROS and RNS) are magnified in intracellular and extracellular environments. 

Once in excess, reactive species react with lipids, proteins, and nucleic acids, altering structural and functional properties of target molecules, thus creating an extensive tissue dysfunction and injury. Every time this noxious impulse becomes a prevailing situation, it drives the human organism to a homeostatic unbalance, between reactive species developed to address detrimental stimulation and the ability that the biological system, has to produce the antioxidants to counteract the excessive damage of reactive species. Therefore, the organism goes into an oxidative stress condition [130].

Since such substantial augment of oxidative species is closely related to diverse inflammatory events on various tissues and organs pathologies, the quantification of those reactive compounds among human fluids becomes in this way pointers of several health disorders. Hence, some of ROS/RNS may be therefore used either as biomarker metabolites of chronic and non-communicable diseases [112,131].

## 13. Oxidative Burst

Considering that the level of mitochondrial ROS generation may be intensified by the increased rate of oxygen use during aerobic metabolism associated to the OP process, whenever an individual experience an exhausting physical activity, or sustains highly caloric (energy-rich) diets, an extra dose of ATP energy will be released by the cells.

Alternatively, whenever the eukaryotic organism underlies an inflammatory noxious stimulus, several innate system mechanisms are activated. In particular:ATP energy for cellular functions extra functions mechanisms (resulted from the extra OP)Signalling messenger mechanism (through the generation of certain ROS)Cytotoxic substances or pathogen invaders combat mechanism (due to oxidative burst during phagocytosis process)

Over a pathologic situation, ROS extra generation demand may be a consequence of:-The necessity to re-establish the energy required (ATP release) to be used by cellular extra work during an inflammatory stimuli condition,-The creation of reactive species to be used as a valuable tool in the defence of the human body, against opportunistic pathogens,-OP intensification, due to the presence of a higher amount of O_2_ during its retake on reperfusion (the so-called ischemia-reperfusion situation).

Upon the recognition of a pathogen microorganism invasion, the human body’s immune system activates the phagocytosis process that makes use of the oxidative burst mechanism during mass consumption of O_2_, to produce high soars of required ROS to be used in its defence. In this particular case ROS specifically created during the oxidative burst process (Figure 9); have shown to be helpful tools/means for the destruction and defence of the organism, against microbes.

Besides, the cells used in immune system defence, express several pattern recognition receptors [132] that when activated, release various types of inflammatory messengers (cytokines and chemokines), and therefore contributes to the immune system signalling pathway as it attracts other immune cells to the inflammation site and contributes to clear the invaders.

Herein, ROS are released from several cells involved in the immune system, such as the polymorphonuclear leukocytes (PMN) or neutrophils, lymphocytes, monocytes, and macrophages [133]. For the sake of clarity perception, cellular components of the immune system shall be known as PMN cells. These PMN cells, when activated, use various types of methods to strike against pathogens responsible for the inflammatory stimuli.

Membrane-bounded nicotinamide adenine dinucleotide phosphate, NADPH oxidase complex (NOX), located at this type of PMN cellular’ membranes has a primary role as a catalyst in the phagocytosis process. To this extent, NADPH is activated and behaves as a reducing agent of the cytosolic O_2_, leading to the generation of${\;O}_{2}^{●-}$as primary ROS product to be used in the strike of captured pathogens within the phagolysosome chamber (cytoplasmic vesicle formed within a PMN cell) [134,135].

On the other hand, a significant amount of another powerful damaging oxidant is developed among neutrophils granulocytes PMN cells. Accordingly, given the higher concentration of the heme-based *Myeloperoxidase* (MPO), a large amount of H_2_O_2_ is catalytically converted into strong two-electron oxidant HOCl, according to the R5 (already described as one of the subsequent redox reactions of generated H_2_O_2_).

Moreover, free radical nitric oxide ${(\mathrm{NO}}^{●})\;$from macrophages, reacts with${\;O}_{2}^{●-}$, in the presence of H_2_O_2_ yields the highly reactive singlet oxygen $\Delta^{1}O_{2}$ as a very effective ROS in pathogens’ spoilage, according to the mechanism pathway that follows R4, with R22:(R22)$${\mathrm{ONOO}}^{-}+\;H_{2}O_{2}\overset{}{\to }\;.^{1}O_{2}$$

In general terms, during the phagocytose oxidative burst, activated PMN cells release${\;O}_{2}^{●-}$, H_2_O_2_, ONOO^−^, O●H, HOCl, and ${}^{1}O_{2}$, within phagolysosome (cytoplasmic vesicle formed within a PMN cell), to kill invading pathogenic bacteria (that will be used to kill the pathogens harvested by phagocytes.)

Therefore, highly reactive ROS molecules are ubiquitous species permanently formed (routinely generated) overall human body. Therefore, becoming a permanent threat since it may affect the structure of various cellular constituents, such as proteins, lipids, and lipoproteins. Resulting in the mutation and cellular damage, which leads to deterioration of mitochondrial enzyme complexes, membranes, and other structural components of cell microarchitecture, either by direct contact or as a consequence of lipid peroxidation propagation.

If not restrained, these oxidative damage reactions can cause injuries on organelles and cellular structures, such as the disruption of the lipid bilayer cellular membrane, the inhibition of the catalytic activity of the enzymes, or by tampering the message enrolled in nucleic acids.

Therefore, to neutralize these detrimental effects, the human organism has several mechanisms to counteract the oxidative/nitrosative action of free radicals and satisfy their demand for a negative charge.

These antioxidant compounds are stable molecules able to supply electrons to the transient oxidative species such as free radicals.

## 14. Antioxidants 

To minimize ROS detrimental effects, living organisms developed several mechanisms to counteract oxidative/nitrosative action of free radicals, by fulfilling their demand for a negative charge through electron donation. Therefore, eukaryotic cells also produce some biomolecules that delay or prevent the oxidative process by inhibit some oxidations of important cellular substrates and are therefore so-called antioxidants.

Such antioxidants are stable small molecules, owing to the ability to react with both ROS and their precursors, to avoid their formation or to supply electrons enough for ROS intermediates, thereby reducing or neutralizing their reactivity and consequent damages in cellular substrates. 

Therefore, there are two effective strategies used by antioxidants against oxidative damage:
A.Prevention of the generation of ROS
aBonding Inactivation: through binding/inactivation of metalloproteins used in the catalysis of OH● formation.B.Inactivation the reactivity of formed ROS
aVia the enzymatic neutralization of ROS: channelling ROS to specific harmless reactions – usually made by superoxide dismutase (SOD), catalase (CAT), and glutathione peroxidase (GPx).bBy scavenging available ROS: through sacrificial interactions of expendable substrates with ROS –Usually through several vitamins such as ascorbic acid (vitamin C), *α*-tocopherol (vitamin E), *β*-carotene (vitamin A), glutathione, uric acid, and bilirubin.


In some cases, it is difficult to counteract the high reactivity of reactive species formed, and the most effective strategy used against its oxidative damage, being essential the antioxidant processes that prevent their development.

Effectively, the ROS antioxidant system initially arises with the inactivation of the reactive ROS: Specially the transient free radical species. This neutralization is mainly carried on, through elementary protein structures endogenously developed, which have an antioxidant behaviour, such as the superoxide dismutase (SOD), catalase (CAT), glutathione peroxidase (GPx).

As described by RIII, SOD is considered an antioxidant protein as it converts the highly reactive ${\;O}_{2}^{●-}\;$into H_2_O_2_ which is sensitive to a lesser degree (lower reactive).

Although not so reactive itself, H_2_O_2_ under the catalytic effect of several enzymes might react with other species present, giving rise to other oxidative species. Therefore, it is of most importance, the removal of H_2_O_2_ through enzymatic neutralization strategy of catalase (CAT) or the Selenium-based glutathione peroxidase GPx, into water and oxygen (R6, R7 and R8).

Foregoing preventive mechanisms are the most effective in reducing oxidative damage from O●H, albeit not effective. Therefore, some of the ROS may escape from enzymatic inactivation, to initiate the peroxidation of cellular substrates like for instance, the polyunsaturated fatty acids. In this circumstance, the reactivity of the ROS can be constrained by using scavenging systems.

Most of ROS scavenger’s compounds are vitamins, which are exogenously supplied through diet or food supplements, and that sacrificial interact with the oxidized substrate or afford the required electrons to ROS. Usually, those exogenous antioxidants are non-protein based compounds that act against the oxidative damage, according to the following four mechanisms of action:Protein binding inactivation:By the sequestration of transition metal ions.Sacrificial interaction:Through the scavenging and extinction of ROS and/or RNS.Peroxidation chain-terminator:End-up the free-radical chain reactions (for instance, the *α*-tocopherol).Through the retrieval of molecular damage:Mainly to retrieve DNA sequences.


For instance, the peroxidation-closure mechanism (termination-step) is crucial to hinder the propagation of the oxidative chain of polyunsaturated fatty acids (PUFA), thereby interrupting the oxidative breakdown of membrane cells.

This suspending mechanism is relevant for the correct functionality of the cellular membrane, as it is damaging affect its fluidity, affecting the diffusion of proteins and other biomolecules within the membrane and consequently their functions amongst the cellular structures.

Note that, most of the oxidative disruption of biological phospholipids occurs on cellular membranes including mitochondria, peroxisomes, and plasma membranes.

Such a procedure involves the development of several small molecules able to supply electrons to unstable ROS species, so they become non-reactive, designated by scavengers.

This electronic compensatory mechanism may involve several pathways:Hydrogen atom transfer,Single-electron transfer,Antioxidant’s ability to chelate transition metals.

Amongst exogenous antioxidants, some are readily available through the dietary intake; others must be synthesised due to their excessive demand or low-level dietary supply.

Each antioxidant plays a role in a specific part of the human organism and operates in a specific environment. Effectively some antioxidants are water-soluble, and others just functionalities in lipid surroundings (illustration of the several antioxidant types *–* shown in Figure 10).

Antioxidant species aims to achieve a balance between pro-oxidant reactive species and antioxidant agents to compensate for the formation thereof and maintain homeostatic balance.

## 15. ROS as Signalling Messengers of Human Health Status 

Generally, at a normal physiological state, ROS are in a homeostatic basal amount, due to the counter-balanced action of commonly created antioxidant species that inhibit the oxidative action of the ROS. In this state, ROS is relevant to support fundamental biological activities such as energy production, cell growth regulation, and intercellular signalling. 

Among such physiological state, generated H_2_O_2_ is one of the main signalling agents when is found at low levels in the human body [136].

Nevertheless, in a pathological condition, generated ROS are present in substantial amounts within the organism due:Over-production of ROS, resulting from the higher generation due to an inflammatory stimulus.Antioxidant system disability to exclude the inherent reactivity of ROS at the same rate at which these species are produced.

The continuous exacerbating production of ROS in the body, along with the impairment of inhibitory capacity from antioxidant available, impels the magnification of the oxidation damage reactions on human tissues and carries some amendments in various signalling pathways, therefore creating an oxidative stress condition.

Such a stress condition, which in turn triggers an inflammatory response, entails the production of more ROS, in a cycling process.

One outcome of oxidative stress is the amendment of the structure and operation of cellular proteins and lipids, which entails several damaging effects for cellular functionality, such as cells membranes’ structural damage, the impairment of cell transport mechanisms, and energy metabolism, the disruption of enzymatic activity, the alteration of cell cycle control and signalling pathway. During a pathophysiological state, there is an overall dysfunctional biological activity supplemented by immune activation and inflammation.

In an oxidative stress/pathological state, ROS acquire new functionalities: the pathological status signalization. Accordingly, some ROS (mainly $O_{2}^{●-}$ and H_2_O_2_), serve as secondary messengers of the pathological stage, by signalling various molecules of the immune system.

The human organism employs many physiological adaptations to respond to changing environmental readily, and therefore maintain the homeostasis. Thus, anytime the biological system of the human organism undergoes a perturbation from its normal physiological state, as a response, it tends to adapt to new demands settled by the adverse conditions. This cell adaptation bias may lead to long-term inflammation processes that may overwhelm the normal function of cells and tissue biology.

Therefore, severe or long-termed oxidative-stress conditions may trigger abnormal mutations on cells and tissue biology, and some inflammation processes may become chronic and accountable for the development of several adverse health status, such as the common NCDs diabetes, CV disease, and cancer [17].

In general, long-term inflammation processes drive most of the NCDs, and many of them, share similar molecular alterations, and common pathophysiological mechanisms among different organs [137].

Low levels of inflammation are set by the body’s perception of an internal threat, even when there is no disease to fight or injury to cure. This awareness (with or without cause) sometimes signals the immune system’s response. Chronic systemic inflammation corresponds to low-grade inflammation stimuli throughout the body, which results in persistently elevated concentrations of circulating pro-inflammatory cytokines, across the life span, which has been associated with the development of both age and life-habits-related NCDs [134]. Many NCDs are, to a large extent, preventable and modifiable risk factors related to lifestyle. 

Low-grade inflammation (a persistent low-level of inflammation) is renowned for playing a role in the development of numerous NCD, including obesity, diabetes, CV diseases, several cancers, respiratory disorders, some impaired neurodevelopment, and mental health disturbances.

These NCDs ailments are at close range related to several biochemical compounds likewise found in several different health disorders. This provides scope for different diseases to interact with each other, affecting the course and outcome of each illness, or even the simultaneous or sequential existence of several health disorders in the same individual/person. Conducting a comorbidity state [138].

The kind of disease-related metabolism of an individual is reflected by some changes in their metabolites (i.e., amino acids, lipids, and organic acids) within body fluids, organs, and/or tissues. Therefore, such metabolites alterations become markers of the disorder’s onset, progression, and prognosis.

Thereafter, it will be considered some metabolites, whose variability in its concentration among human fluids is deeply related with several of the most common NCD; mainly the obesity, diabetes *mellitus*, cardiovascular disorders, chronic obstructive respiratory, neurodegenerative diseases and ultimately cancer, and their relationship with hydrogen peroxide, ascorbic acid, uric acid, and dopamine.

### 15.1. Obesity

Obesity is an international public health concern with a significant impact on the entire developed world [139]. It arises from the nutritional stress, caused by an excess of high-fat and/or carbohydrate diets, which creates an energy imbalance, in which the intake of energy surpasses its expenditure [140].

This imbalance can be triggered either by the food type and/or the amount consumed, which promotes oxidative stress within the human organism, as it is evidenced by the increased lipid peroxidation products, protein carbonylation, and also the decreased antioxidant status of the body [141].

Bad-diet habits over long periods may trigger chronic activation of the innate immune system that promotes a low-grade systemic inflammation state of the organism, which in turn induces the development of several pathologies, such as the impaired signalling and the dysregulation of metabolic pathways [125] such as the development of insulin resistance and T2DM.

Overweight metabolic disruption is highly related to high blood pressure [142] (hypertension) among obese individuals and considered the number one risk factor for stroke, congested heart failure (CHF), coronary heart disease (CHD), ventricular dysfunction, and cardiac arrhythmias [143].

#### 15.1.1. Relation with Amounts of Uric Acid

Uric acid (UA) is the final enzymatic waste product created during the normal breakdown of purine nucleosides in humans and higher-order primates. In such catabolism, nitrogen bases undergo deamination (withdrawal of amino group) in xanthine, and then the resulting carbon skeleton experience a disintegration into uric acid (in two tautomeric forms) as represented in Figure 11. 

Most other lower animal species may further process uric acid to yield more-soluble end products since they bear an essential enzyme responsible for the additional oxidation of uric acid into allantoin; the uricase [144]. Accordingly, uric acid levels may vary in human populations as is dependent on external factors such as diet, and excess intake of foods high in purine content or a deficiency of uricase supplements (given the human lack of uricase gene to convert uric acid into its soluble form [141] of allantoin), or even due to genetic factors [145].

As previously referred, UA is an effective suppressant of reactive species in the extracellular medium, and the most abundant free radical scavenger, which accounts for up to 60% of plasma antioxidative capacity protection of cells from oxidative damage. Therefore, during its disposal out of the blood by kidney-filtering, the human kidney reabsorbs ~90% of the filtered urate into the blood in the proximal tubule of the nephron and reclaims part of the UA [146,147], and only about ~10% of UA is discharged by the urinary system.

The average concentration of UA in human blood is close to its water solubility limit (~6.8 mg dL^−1^), while the reference ranges for serum UA are 3.5 to 7.2 mg dL^−1^ in adult males and postmenopausal women, and 2.6–6.0 mg dL^−1^ in premenopausal females [148]. Whenever UA in the blood is above 7 mg/dL, whether by urate ion overproduction or due to its’ poor urinary excretion, the condition of hyperuricemia occurs. Under higher levels, the urate ion (mono-ion) ensued uric acid, exhibit a preference for precipitate along with sodium ions, as monosodium urate salt (MSU), as depicted in Figure 12 [147].

Under a hyperuricemia condition, the crystals of urate salt formed, tends to accumulate in various specific areas of the body. Mostly in the kidneys (in the form of urate stones), ligaments, and joints (covering the hyaline cartilage of joints with MSU needle-like crystals), resulting in the emergence of characteristic diseases, such as gouty arthritis and stones in the kidneys.

Additionally, hyperuricemia is often associated with patients with obesity and metabolic syndrome [143], abnormal cholesterol or triglyceride levels, excessive abdominal fat, and high blood glucose levels, and frequently engaged with vascular endothelial impairments like hypertension [149] and corresponding cardiovascular disorders.

All disturbances that result from high levels of UA, along with the origin overweight associated, drives to a condition of impaired mobility and flexibility collapse that hasten the aging process. Consequently, it is of utmost importance the accurate detection and monitoring of UA levels in human fluids trace samples.

#### 15.1.2. Relation with Amounts of Dopamine

The lateral hypothalamus is the prime region of the brain regulating food intake and nutrition requirements [150], where dopaminergic mechanisms modulate feeding behaviour. During a reward stimulus, like food intake, dopaminergic mesolimbic pathway connects the main dopamine producing-area: the ventral tegmental area, localized in the midbrain, to the ventral striatum of the basal ganglia in the forebrain, where are confined several areas associated with satisfaction (hippocampus, amygdala, prefrontal cortex, and nucleus accumbens). Biologically, dopaminergic neurons formed from tyrosine in the ventral tegmental area, are projected to the nucleus accumbens of the dorsal striatum area (Figure 13). From a physiological point of view, this dopamine mesolimbic system plays a key role in emotional, motivation and is strongly associated with reward symptoms.

Dopamine is an important endogenous catecholamine that functions as a neurotransmitter and hormone within the body.

When exposed to a rewarding stimulus like food, the brain responds by increasing the release and activation of dopamine neurons (the neurotransmitter associated with pleasure and satisfaction) [143,144]. When an individual anticipates experiencing something rewarding [152], dopaminergic neurons are activated and projected into the area of the acumen’s nucleus, thus increasing dopamine levels in this area. This build-up of dopaminergic neurons stimulates the repetition of the pleasant action previously encountered. Therefore, dopamine molecule is recognized to be a key agent for food reward and food intake control.

Nevertheless, repeated exposure to a food reward, as is the case of overeating, contributes to a reduction in dopaminergic function at the dorsal striatum region, along with the dopamine D2 receptor sensitivity at the nucleus accumbens, this way decreasing the sensitivity to the rewarding effects of food consumption. This decline in the activity of dopaminergic neurons among the nucleus accumbens area can trigger a compensatory over-eating behaviour, which leads to the increase of the reward/pleasure experience, this way developing the obesity [143,144].

### 15.2. Diabetes Mellitus

The global incidence of diabetes is currently reaching pandemic proportions worldwide, with type 2 diabetes *mellitus* (T2DM) as the most prevalent and serious metabolic disease all over the world. 

T2DM is a metabolic disorder characterized by high blood glucose levels and insulin resistance, characterized by a pancreatic β-cell dysfunction, in which oxidative stress is thought to be a primary cause. 

It is observed that long-stand periods of high blood-glucose levels is generally linked to both macro and microvascular complications, such as CV diseases, strokes, atherosclerosis, neuropathy, retinopathy, and nephropathy [153]. This outcome from the association of oxidative stress and chronic inflammation to the development of T2DM metabolic disorder.

Effectively, high levels of intracellular glucose result in higher tricarboxylic acid yield to serve in the Krebs’ cycle as an electron donor. This leads to an increase in the mitochondrial proton gradient during oxidative phosphorylation, which increases the lifetime of ETC’s electron carriers. Consequently, increasing the amounts of leaked electrons, which will combine with respiratory oxygen and give rise to more ROS species, especially superoxide anion.

The hyperglycaemia status is triggered by the persistently high level of glucose in the blood and is responsible for the subsequent augmentation of ROS give rise to *β*-cell dysfunction and increase insulin resistance, which leads to a deterioration of T2DM condition [132,154].

In a systemic insulin resistance state, the insulin signalling within glucose recipient tissues is therefore defective, and in this case, hyperglycaemia perseveres. Inadequate glucose sensing to stimulate insulin secretion [133] results in the prevalence of elevated glucose concentrations in blood and to *β*-cell dysfunction. Both pathological states concerning *β*-cells dysfunction and insulin resistance, influences, and synergistically exacerbates T2DM [132].

#### 15.2.1. Type 2 Diabetes Mellitus Associated Disorders

T2DM condition is liable for a high level of glucose in the blood, which is closely related to endothelial cells dysfunction. The vascular endothelium is represented by a monolayer of endothelial cells through the internal surface of the blood vessels.

This vascular endothelium defectiveness arises from abnormalities in the production or bioavailability of endothelial-derived nitric oxide, which is responsible for posterior/subsequently deleterious variations in vascular reactivity [155], like the atherosclerotic plaque build-up inside arteries.

Overtime [156], such plaques get harden and narrows the vessel arteries, which trigger a hypoxia condition, due to impaired oxygen-rich blood flow, leading to hazardous effects on organs structure and function.

This limitation of oxygen-rich blood flow is the main cause of some CV diseases, like ischemic heart disease, also called coronary heart disease (CHD) or coronary artery disease (CAD). Depending on the affected organ, this ischemic status has other designations, namely, when there is cerebral ischemia, in which poor oxygenated blood flow to the brain, is assigned as stroke, whenever the ischemia occurs in myocardial muscular tissue; it is identified as heart attack [143].

Therefore, T2DM is one of the metabolic syndromes, responsible for the increased risk of developing one of the world’s major non-communicable diseases, coronary heart disease (CHD), and CV disease [157].

Besides high blood glucose levels, all the other metabolic syndromes trigger the inflammatory condition that is responsible for several health diseases. These include; increased blood pressure, excess body fat around the waist, and abnormal cholesterol or triglyceride levels [158].

Therefore, CV diseases result from chronic inflammatory conditions on the arterial vasculature caused by endothelial cells dysfunction [159]. Namely due to vessel atherosclerotic lesions, using narrowing of the arteries caused by a build-up of plaque, thus contributing to its pathogenic state [147].

#### 15.2.2. Relation with Amounts of Hydrogen Peroxide:

High levels of glucose in blood results in a higher oxidative phosphorylation mitochondrial metabolism with the formation of more substantial amounts of mROS.

Under glucose stimulation, endocrine β-cells generate higher amounts of mitochondrial $O_{2}^{●-}$ that through the action of metalloenzyme MnSOD converts the $O_{2}^{●-}$ into H_2_O_2_ and O_2_, among the mitochondrial matrix region. At physiological conditions, such mitochondrial H_2_O_2_ exists in low concentration amounts (low levels) and under glucose stimulation, acts as a signalling mediating agent in β-cells insulin secretion, commanding pancreatic islet β-cells to secrete insulin into the bloodstream [160]. Therefore, glucose increases the level of intracellular H_2_O_2_, which in turn alters the insulin secretion rate in pancreatic β-cells.

However, long-term blood exposure to glucose conducts a high metabolic ETC flux that generates harmful levels of mROS like the higher amounts of H_2_O_2_ that has both mediating and toxic effects [161]. At higher levels, H_2_O_2_ may drive an oxidative stress prevailing situation, which in turn becomes harmful to pancreatic islets, as it might lead to a decrease in *β* -cells’ mass (might lead to the loss of *β*-cell’ mass) [135].

During high blood glucose levels (hyperglycaemia) timer-span exposure, there is a high metabolic generation of harmful levels of mROS, and consequently, the amounts of H_2_O_2_. In turn, this might lead to a decrease in *β*-cells’ mass which results in its dysfunction with low insulin gene expression (as shown in Figure 14), resulting in lower insulin secretion and insulin resistance, increasing the impairment of T2DM condition [134].

#### 15.2.3. Relation with Amounts of Ascorbic Acid

Vitamin C or ascorbic acid (AA) is an essential nutrient with high antioxidant potential, obtained from the diet, whose structure is quite similar to carbohydrates (such as glucose) [162].

Given such resemblance (Table 3), dehydroascorbic acid (DHA) (oxidized form of AA) competes with glucose molecule for its transport towards the cell and penetration into the cell membrane, through protein transport involved in glucose translocation across the cell membrane by the glucose transporters (GLUT) [163].

Pancreatic *β*-cells are accountable for the secretion of the hormone that establishes the transport of glucose into the cell and this way largely involved in regulating blood glucose levels, the insulin [144]. Such *β*-cells tend to exhibit low levels of antioxidant enzymes, thereby becoming more susceptible to oxidative stress [164] and pancreatic *β*-cell dysfunction.

Consequently, antioxidant properties that distinguish AA are of great importance for proper secretion of insulin by β-cells in pancreatic islets, given that it mitigates oxidative damage caused by the large amounts of ROS during oxidative stress.

In a diabetic condition, monocyte cells are recruited into the vessel walls, where NADPH oxidase is activated, thereby contributing to increase the production of superoxide anion, and ROS augmentation. This increment in ROS production promotes the inflammatory state of the vascular endothelium tissue, an associated complication of T2DM condition [165].

Therefore, AA inlet relieves in some extend the symptoms associated with several T2DM complications, by decreasing the dysfunction of β-cells in pancreatic islets, associated with T2DM patients [166]. High dosages of AA have proven to reduce several complications associated with T2DM, especially microvascular complications, such as retinopathy, nephropathy, and diabetic foot [144].

### 15.3. Chronic Obstructive Pulmonary (COP) Diseases

Chronic obstructive pulmonary disease [167] consists of a chronic inflammatory response in the airways to noxious particles and gases, especially on bronchial tubes that become narrowed and covered on mucus. This airways infection is usually a progressive condition that entails a persistent airflow limitation and consequent defective exchange between O_2_/CO_2_. Such bronchial mucus hypersecretion leads to an increased presence of bacteria, which further increments airway inflammation.

#### Relation with Amounts of Hydrogen Peroxide (H_2_O_2_)

Airway inflammation [168] in chronic obstructive pulmonary disease is closely correlated with the oxidative stress involved in such pathogenesis. This exacerbation of oxidative species is responsible for lung tissue injury, damage and apoptosis of the airway epithelial cells, the obstruction of airflow, and in the development of bronchial mucus. Besides, oxidative stress exhibits pro-inflammatory effects that contribute to the inactivation of antioxidant enzymes.

Given the fact that H_2_O_2_ is a mediator of oxidative stress, it plays an important role in chronic obstructive pulmonary disease.

For instance, in 2013, Kazuya Murata et al. conducted a non-invasive lower respiratory tract biological sampling study (through expired breath condensate collection), where high levels of H_2_O_2_ were identified in patients with advanced or severe chronic obstructive pulmonary disease. A high concentration of H_2_O_2_ in the airways is thus indicative of the increased oxidative stress and inflammatory process in the airways. As such, H_2_O_2_ levels of expired breath condensate are regarded as useful biomarkers of oxidative stress for the treatment of associated chronic obstructive pulmonary disease. In Murata study, significant correlations between H_2_O_2_ concentration and the number of eosinophils and neutrophils during induced sputum were also reported.

Therefore [167], H_2_O_2_ levels obtained in exhaled breath condensate biological samples may represent a quantitative evaluation of airways oxidative stress/inflammation, becoming a useful marker in the assessment of the symptomatic severity of the chronic obstructive pulmonary disease.

### 15.4. Neurodegenerative Diseases

Neurodegenerative disorders are normally associated with a progressive loss of neuron cell activity, and consequent deficit in specific brain function, performed by certain affected central nervous system regions. 

Such a brain functioning disruption could be related to a memory deficit, defective movement, and impaired cognition. This could be associated to massive death of dopaminergic neurons [161] in the substantia *nigra* region of the brain, like Parkinson’s disease, or either on account of the aggregation [169] of amyloid (A*β*) protein agglomerates in senile plaques, such in case of Alzheimer’s disease.

The progression of such neurodegenerative disorders is triggered by an exacerbated production of ROS and is believed that they play a critical role in neurodegenerative diseases since high levels of oxidative stress are commonly observed in the brain of patients with neurodegenerative impairments [161].

Owing to the weak antioxidant defence, high oxygen consumption, and the enrichment in polyunsaturated fatty acid content the central nervous system is particularly sensitive to oxidative stress. Especially neurological cells are particularly vulnerable to the oxidative damage caused by exacerbate amount of ROS during oxidative stress [170].

### 15.5. Oxidative Stress on the Brain

Apart from the human body, the inflammatory process affects the human brain, owing to the exacerbating presence of ROS when under a pathological condition [171].

Having a major energy-demanding, brain cells require a substantial amount of oxygen, to perform intense metabolic activities concerning brain functions, with metabolic activities generating numerous types of ROS that in a physiological condition, are develop as support media in the brain cell growth, neuroplasticity, and cognitive functioning [172]. Nevertheless, ROS generated in the brain may potentially undergo oxidative stress, making brain cells particularly vulnerable to molecular damage as well. As a result, oxidative stress also contributes to several neurodegenerative conditions, such as Alzheimer’s disease and Parkinson’s disease [100].

#### The Amount of H_2_O_2_ in Neurodegenerative Disorders such as Parkinson and Alzheimer’s Pathologies

By its good functioning, the human brain is the organ that requires high-energy yield in the form of ATP to be used by neuron cells. Thereby, owing to brain O_2_ large consumption during its functioning, and neuronal membranes enrichment in PUFAs, the central nervous system is particularly prone to oxidative phosphorylation, which makes it sensitive to the oxidative stress, and particularly vulnerable to lipid peroxidation [173].

An increasing amount of pieces of evidence implies the involvement of lipid peroxidation and nucleic acid oxidation in the development and progression of several mammalian brain neurodegeneration [170,174,175,176,177].

Lipids typically found in large amounts in plasma membranes are more prone to oxidizing effects of reactive species such as ROS, and thus suffer oxidative deterioration (lipid peroxidation). As a result, membrane-bounded enzymes are impaired, engendering their inactivation, and the fluidity of the membrane decreases allowing the passage of previously blocked species, and thus the structure and function of the cell membrane are affected. By disrupting the function of cellular membranes, ROS significantly deteriorates neuronal cells.

Oxidative stress might spark several biochemical amendments in biomolecular components, resulting in signalling pathways impairments and metabolism mismatches that urge peptide clusters deposition and breakdown of neurological signalling as well. Although presently there is still no effective means of healing the several neurodegenerative disorders, the understanding of ROS/H_2_O_2_-related mechanisms in disease progression provides important insight to conceived possible therapies/treatments that alleviate associated symptoms.

## 16. Conclusions

Oxidative damage is closely related to impaired energy metabolism, where minor-stresses that mildly impair oxidative phosphorylation may initially cause slight increases in ROS. These ROS can then further impair oxidative phosphorylation and thus lead to an even greater release of ROS, in a cyclic process. Early disease detection methods are based on the recognition of a minimal change-level of a disease-specific biomarker in body fluids.

Biomolecules concentration amendment quantification is relevant to recognize both normal biological cell signalling and pathological health conditions. The level of those compounds in biological fluids may be used as a potential biomarker of whole body pathophysiologic state) [178].

In the future, the development of new non-invasive and reliable electrochemical methodologies is recommended, allowing the detection and quantification of small variations in the concentration of biomarkers that allow the early diagnosis and adequate monitoring of NCDs.

## Figures and Tables

**Figure 1 biosensors-10-00121-f001:**
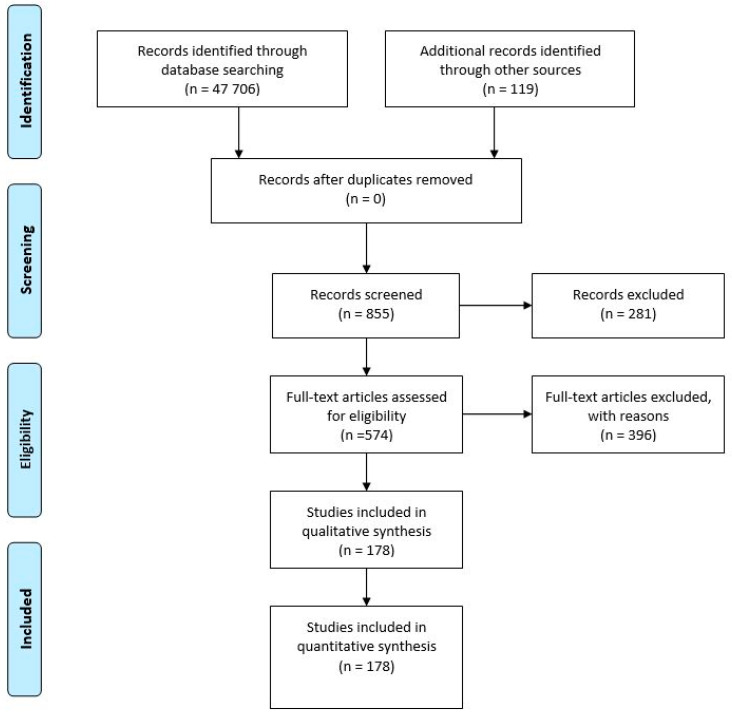
PRISMA flowchart with the global results of the literature review and applied screening.

**Figure 2 biosensors-10-00121-f002:**
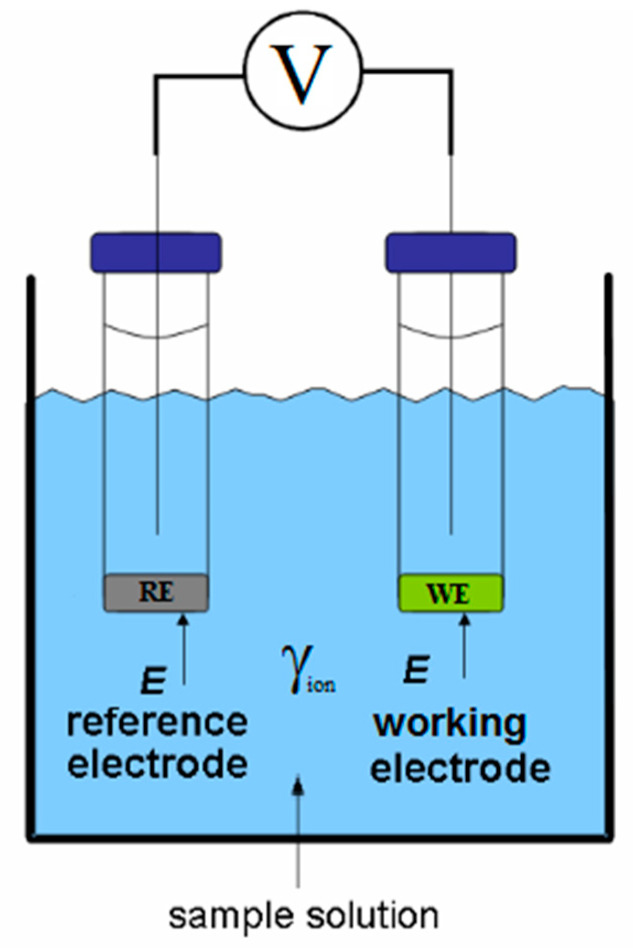
Setup of the potentiometric biosensing measurement.

**Figure 3 biosensors-10-00121-f003:**
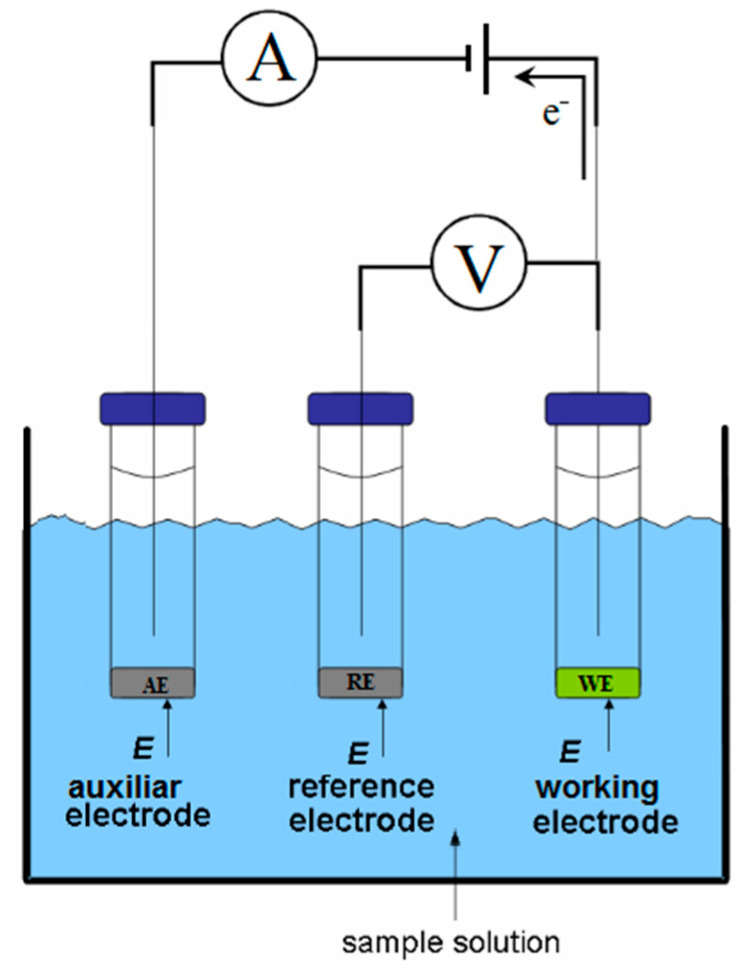
Setup of an amperometric sensor, where is applied to a potential difference between the working and reference electrodes, whilst the faradaic electric current is measured as long it flows towards the auxiliary electrode.

**Figure 4 biosensors-10-00121-f004:**
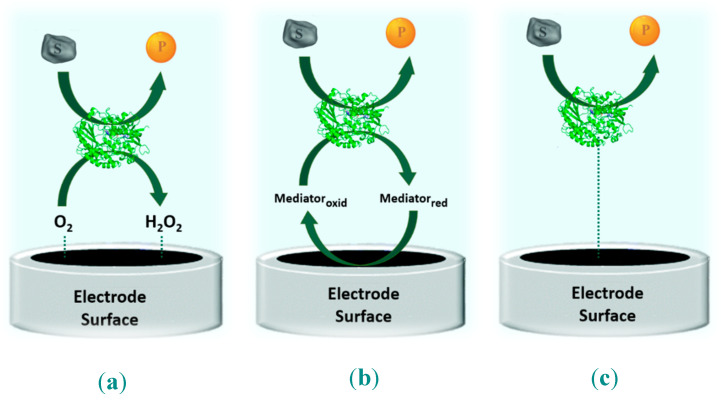
Schematic illustration of (**a**) First-generation amperometric glucose biosensor, with the electrons resulting from *β*-D-glucose substrate (S) enzymatic (GOx) reaction reducing the molecular O_2_ into H_2_O_2_ as produces the gluconolactone (P). (**b**) Second-generation amperometric glucose biosensor, where the electrons resulting from *β*-D-glucose oxidation reaction (S to P) are transferred to a mediator molecule that transports them from GOx enzyme active site to the electrode surface. (**c**) Third-generation amperometric glucose biosensor, where the electrons resulting from *β*-D-glucose oxidation are directly transferred from the redox-active site of enzyme cofactor (GOx) to the electrode surface previously modified. Adapted from [54].

**Figure 5 biosensors-10-00121-f005:**
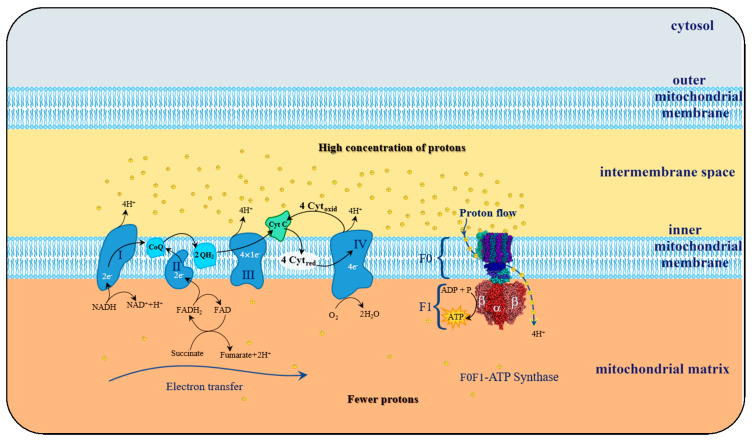
Mitochondrial respiratory metabolism. The electron transport chain has the main purpose of the creation of a transmembrane electrochemical proton gradient that will power the OP and ATP release. Electrons are transferred along the chain of protein complexes I to IV. Complex I-NADH dehydrogenase; complex II-succinate dehydrogenase; complex III-cytochrome c oxidoreductase; complex IV-cytochrome c oxidase; complex V-F1F0 ATP synthase.

**Figure 6 biosensors-10-00121-f006:**
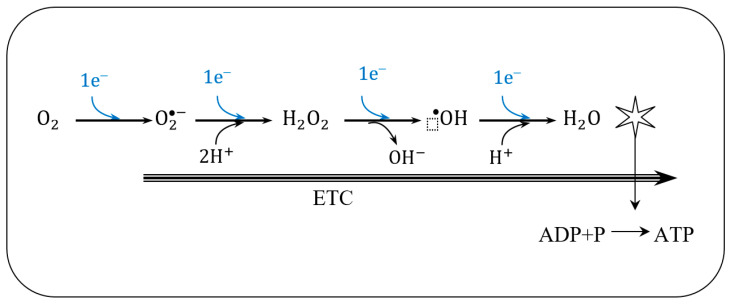
Schematic illustration of - Series of one-electron reaction that occurs during the ETC process.

**Figure 7 biosensors-10-00121-f007:**
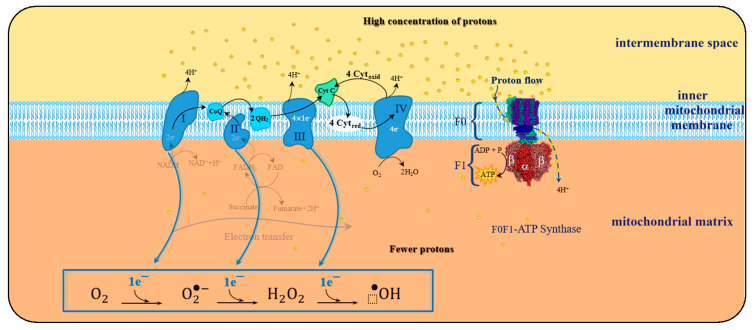
ROS generation by electron leakage during ETC of OP of mitochondrial respiratory metabolism.

**Figure 8 biosensors-10-00121-f008:**
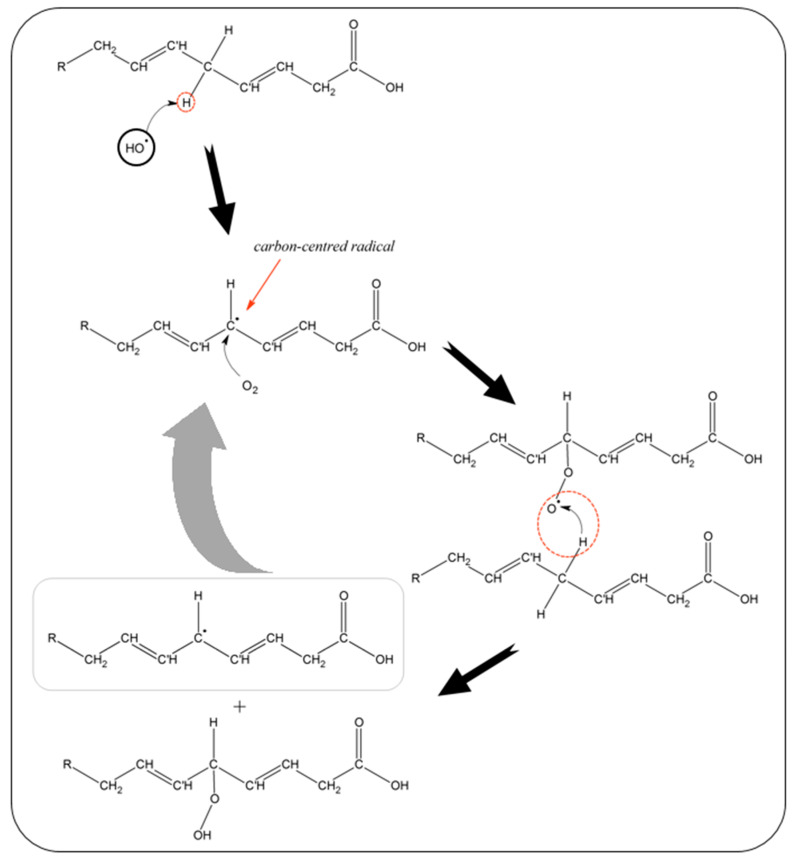
Reactivity of hydroxyl radical (OH●) over lipid chain– resulting in lipid peroxidation.

**Figure 9 biosensors-10-00121-f009:**
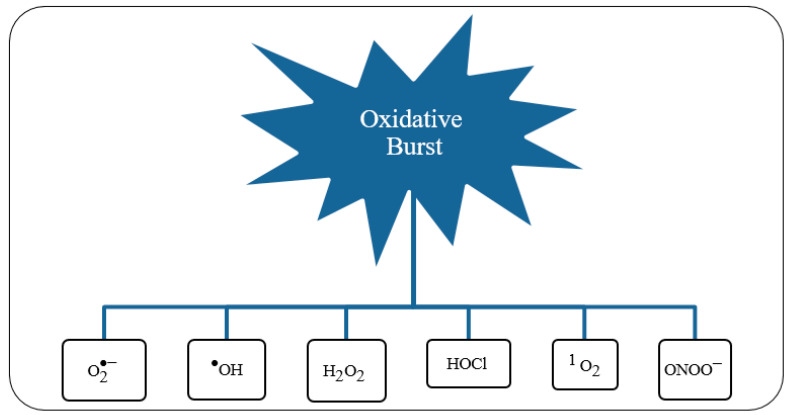
Major ROS formed during the PMN mitochondrial burst.

**Figure 10 biosensors-10-00121-f010:**
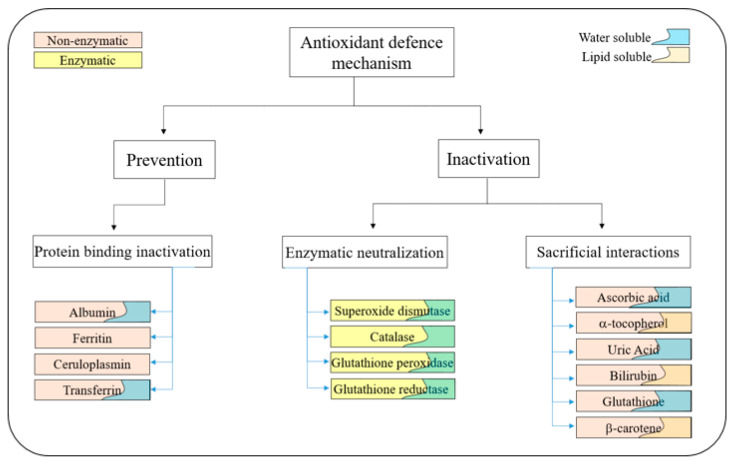
Representation of several types of antioxidants that operate within the eukaryotic organisms.

**Figure 11 biosensors-10-00121-f011:**
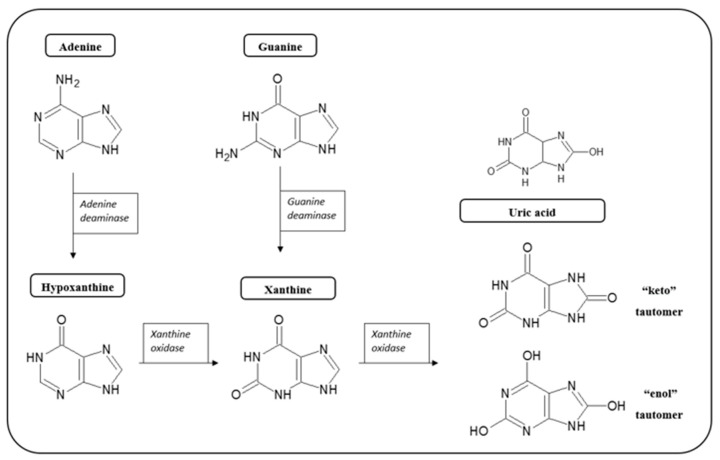
Catabolism of the purines, adenine, and guanine into uric acid.

**Figure 12 biosensors-10-00121-f012:**
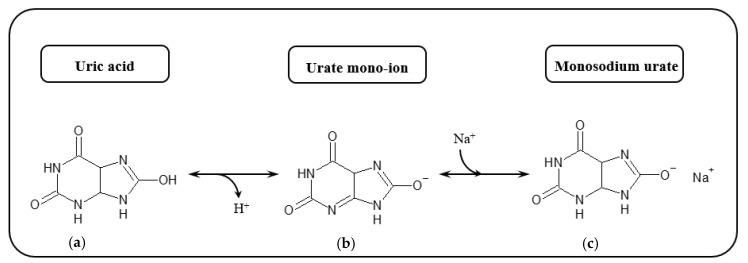
The various chemical structures of uric acid in the physiological medium. (**a**) UA structure after purine breakdown (illustration of UA keto tautomer; (**b**) under physiological pH of 7.4 and 37 °C, UA predominately exists in the form of deprotonated monoionic urate; (**c**) owing to sodium significant concentration and ionic strength, urate ions are disposed to precipitate as monosodium urate salt.

**Figure 13 biosensors-10-00121-f013:**
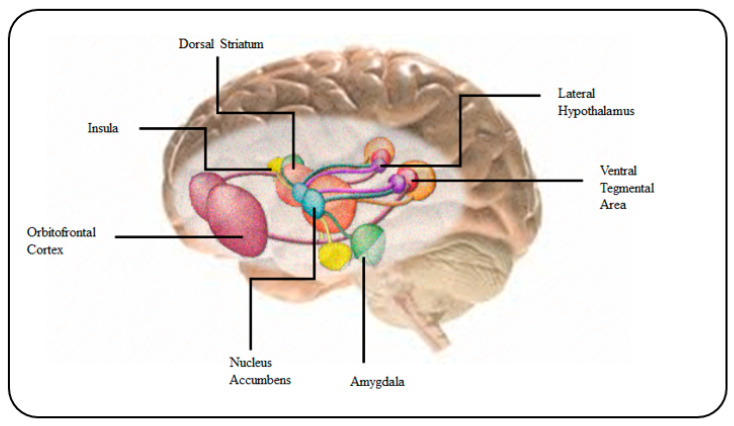
Areas of the human brain implicated in the rewarding effect of food. Adapted from [151].

**Figure 14 biosensors-10-00121-f014:**
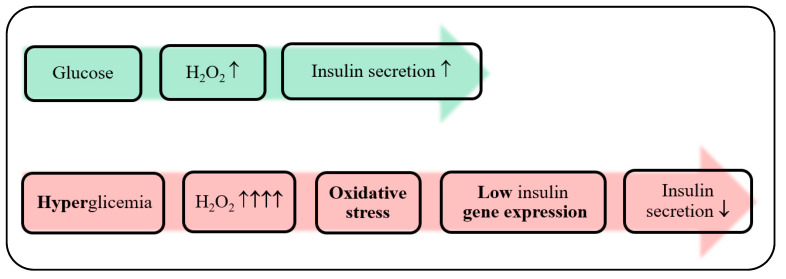
Schematic representation of a model of H_2_O_2_ as a signal in blood glucose-stimulated insulin secretion.

**Table 1 biosensors-10-00121-t001:** Interfacial electrochemical techniques employed in electrochemical biosensor devices.

Method	Potentiometry	Amperometry	Voltammetry	Impedimetry
**Measured Property**	Electrode Potential/*E*	Electric Current/*I*	Electric Current/*I* = *f(t)*	Electrical Impedance/Z
**Controlled Property**	Electric Current/*I* = 0	Electrode Potential/*E*	Electrode Potential/*E* = *f(t)*	Electrode Potential/*E* = const + *f(t)*

**Table 2 biosensors-10-00121-t002:** Lewis structure of molecular oxygen, and fundamental reactive oxygen species formed in the mitochondrial respiratory engine of the cell.

O_2_	^●^O_2_^−^	^●^O_2_^2−^	H_2_O_2_	^●^OH	OH^−^
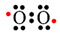	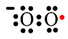	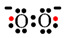	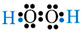	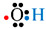	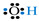
Oxygen	Superoxide anion	Peroxide	Hydrogen peroxide	Hydroxyl radical	Hydroxide anion

**Table 3 biosensors-10-00121-t003:** Similarities between configurations of D-glucose and dehydroascorbate hydrate structure.

D-Glucose	DHA Hydrate
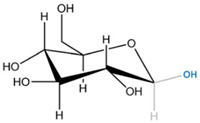	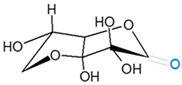

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
