# Peer review of "Biomolecules and Electrochemical Tools in Chronic Non-Communicable Disease Surveillance: A Systematic Review"

_biosensors, 2020, doi:10.3390/bios10090121_

Round 1

Reviewer 1 Report

Biomolecules and electrochemical tools in chronic non-communicable disease surveillance

Ana etal

Authors nicely reviewed electrochemical techniques for biosensor senor and other bio identifications and well documented and following minor revisions are needed to publish above paper in Biosensors

  • The paper was well written, nicely given the introduction
  • Few minor points to added on Impedance spectroscopy , page 12 Line 405 : in addition to 58 ref. electrochemical uses to be further improved on reviews on impedance applications: Materials 13 (8) (2020)1884), and Chemical Reviews 113(2013)5364, Electrochemistry communications 9 (3) (2007)409, Materials Science and Engineering: B 177 (1),(2012) 100
  • Impadence spectroscopy used capacitance (surface film, double layer, bulk ) and resistance (surface film, charge transfer and bulk resistance) and diffusion co-efficients. In addition to refs (31): It will nice to include the few other applications of impedance spectroscopy (Electrochimica Acta 128(2014)198 Journal of Solid State Electrochemistry 16 (5) (2012)1833, Langmuir 34 (5) (2018)1873, J. Physical Chemistry C 117 (18) (2013), 9056, Electrochimica acta 85(2012) 572, J. Electroanalytical Chemistry 603 (2)(2007) 287, J. Solid State Electrochemistry 21 (10) (2017) 2921
  • Please correct few errors in the paper and improve the quality of figs
  • Nicely discussed the overview of nanoparticles as electrocatalysis
  • Few errors in lines 623, 680, 1229 and few other places

Author Response

Journal: Biosensors (ISSN 2079-6374)

Manuscript ID: Biosensors-897685

Type: Review

Title: Biomolecules and electrochemical tools in chronic non-communicable disease surveillance

Reviewer 1

Comment: Authors nicely reviewed electrochemical techniques for biosensor senor and other bio identifications and well documented and following minor revisions are needed to publish above paper in Biosensors

The paper was well written, nicely given the introduction

Answer: We appreciate your kind words as well as the critical and careful reading of our work.

Comment: Few minor points to added on Impedance spectroscopy , page 12 Line 405 : in addition to 58 ref. electrochemical uses to be further improved on reviews on impedance applications: Materials 13 (8) (2020)1884), and Chemical Reviews 113(2013)5364, Electrochemistry communications 9 (3) (2007)409, Materials Science and Engineering: B 177 (1),(2012) 100

Impadence spectroscopy used capacitance (surface film, double layer, bulk ) and resistance (surface film, charge transfer and bulk resistance) and diffusion co-efficients. In addition to refs (31): It will nice to include the few other applications of impedance spectroscopy (Electrochimica Acta 128(2014)198, Journal of Solid State Electrochemistry 16 (5) (2012)1833, Langmuir 34 (5) (2018)1873, J. Physical Chemistry C 117 (18) (2013), 9056, Electrochimica acta 85(2012) 572, J. Electroanalytical Chemistry 603 (2)(2007) 287, J. Solid State Electrochemistry 21 (10) (2017) 2921.

Answer: Recommended works are quite illustrative, about the additional features gauged the techniques of electrochemical impedance spectroscopy. Namely, the diffusion coefficient of ionic species that establish the electrocatalytic materials and resultant intrinsic ionic conductivity of different materials used in electrochemical biosensors' electrode assembly. A fundamental parameter for the evaluation of the catalytic performance of the biosensor.

Moreover, the specific works regarding high voltage electrolyte materials, such as lithium-ion batteries, afford for long term energy supply, which is fundamental for most of the intervenient medical devices.

We are very grateful that you had provided us with the opportunity to get to know some outstanding works, which we have read with great care and attention. We deem it to be of great relevance, and we will certainly be able to add in a future review article on medical devices.

Comment: Please correct few errors in the paper and improve the quality of figs

Nicely discussed the overview of nanoparticles as electrocatalysis

Few errors in lines 623, 680, 1229 and few other places

Answer: We very much appreciate the observations and the identification of the errors, which have already been corrected in the document.

The figures were carefully analysed and those that were not with quality were improved.

Reviewer 2 Report

The manuscript displays a sound and well-constructed review on electrochemical analysis and biosensing of chemical species for disease surveillance. The review is well written, educational and it approaches many different aspects of biosensors and analytical techniques. In some points, the manuscript has abrupt changes from subject to subject, a suggestion for further improvement would be to consider making a better linkage between themes and paragraphs were this abrupt changes of theme occurs. Another revision that I suggest is to write a brief section on the methodology of the review process, displaying: Databases searched, data range(2010-2019), terms used in the search, number of articles retrieved in the initial draft, number of articles filtered, the filter (the reason for the removal of the initial draft of articles). This is a requirement for systematic reviews, because mdpi journals follow the PRISMA guidelines, which is present in the Guide for Authors. I suggest presenting the review information that I stated previously (Just as a reminder, this is a requirement for systematic reviews and has nothing to do with meta-analysis). Nevertheless, the author have produced an article that presents useful informations and performs a good review of biosensing with electrochemical analysis for disease surveillance, that would be interesting for the readers of the journal.

Additional specific suggestions

Introduction

Line 33: Give a quick definition of non-communicable diseases in a shorter manner than it is done in line 52

Line 37: “other related parameters” Give examples

Line 38: “every single medical condition is strongly related and it is enrolled in the fluctuation of several biological factors” Give examples between biological factor and incidence of disease

Line 41-43: Also lacking exemplification

Line 44: “good signal of response” is a vague term

Line 45: The paragraph seems to suggest that a high conductive material leads to analytical selectivity and sensibility, which is not necessarily true. I believe this paragraph should be rephrased in the sense that it should say that: specific materials lead to higher sensitivity / sensibility and why that occurs

Biosensors

Line 152: It would be interesting to see an example

Electrochemical methods employed on disease diagnosis

Line 206: A brief definition of an electroactive compound would improve this section of the discussion

Table 1 should have “Measured / Controlled Property” rather than “Quantity”

Table 1 “Impedimetry” refers to Electrochemical impedance spectroscopy correct? If so the Controlled Property should be the frequency of potential oscillation in a given amplitude. As an example:

freq(Hz)=f(E±E_0 ),being E_0  the amplitude and E the average potential 

Line 369: “small sinusoidal wave” is vague, instead the author may write: “through a sinusoidal wave of potential variation through time”

Line 360-377: this section lack a bit of continuity, the paragraphs cut harshly from subject to subject, I suggest that the authors reformulate each end and beginning of paragraph to make a continuous argument

Line 388: Spectroscopy

Line 390: Give an example of this sensing element

It would be interest to read about at least one example of pathology that may be surveyed with each of the electrochemical techniques: Potentiometry, amperometry, voltammetry and impedance spectroscopy. I suggest that the author write about a brief example of each case, in the pertinent paragraphs

Author Response

Journal: Biosensors (ISSN 2079-6374)

Manuscript ID: Biosensors-897685

Type: Review

Title: Biomolecules and electrochemical tools in chronic non-communicable disease surveillance

Reviewer 2

Comment: The manuscript displays a sound and well-constructed review on electrochemical analysis and biosensing of chemical species for disease surveillance. The review is well written, educational and it approaches many different aspects of biosensors and analytical techniques. In some points, the manuscript has abrupt changes from subject to subject, a suggestion for further improvement would be to consider making a better linkage between themes and paragraphs were those abrupt changes of theme occurs. Another revision that I suggest is to write a brief section on the methodology of the review process, displaying: Databases searched, data range(2010-2019), terms used in the search, number of articles retrieved in the initial draft, number of articles filtered, the filter (the reason for the removal of the initial draft of articles). This is a requirement for systematic reviews, because mdpi journals follow the PRISMA guidelines, which is present in the Guide for Authors. I suggest presenting the review information that I stated previously (Just as a reminder, this is a requirement for systematic reviews and has nothing to do with meta-analysis). Nevertheless, the author has produced an article that presents useful information and performs a good review of biosensing with electrochemical analysis for disease surveillance, that would be interesting for the readers of the journal.

Answer: We appreciate the kindly words stated, as well as the critical and careful reading of our work. The current version of our, work includes a systematic review taking into account the PRISMA guidelines.

Additional specific suggestions

Introduction

Comment: Line 33: Give a quick definition of non-communicable diseases in a shorter manner than it is done in line 52

Answer: We have considered such pertinent indication and hence we have introduced a short definition of NCDs " These NCDs are chronic and non-transferable health conditions; closely …" (Line 34).

Comment: Line 37: “other related parameters” Give examples + Comment: Line 38: “every single medical condition is strongly related, and it is enrolled in the fluctuation of several biological factors” Give examples between biological factor and incidence of disease

Answer:  With the aim clarifying the interconnection between diverse health conditions, along with the imbalance of several biological parameters/factors, it was assign an expressive example that clarifies the association of harmful dietary habits and the development of type-2 diabetes mellitus.

Comment: Line 41-43: Also lacking exemplification

Answer: In order to clarify the relationship between the electroactive recognition elements, and the precise measurement of the target species, some demonstrative examples about the mandatory constraints on biosensors' performance in human body samples, were provided.

Comment: Line 44: “good signal of response” is a vague term

Answer: As recommended, the sentence have been rephrased.

Comment: Line 45: The paragraph seems to suggest that a high conductive material leads to analytical selectivity and sensibility, which is not necessarily true. I believe this paragraph should be rephrased in the sense that it should say that: specific materials lead to higher sensitivity / sensibility and why that occurs

Answer: As suggested, the sentences have been rearranged and the expressions corrected

Biosensors

Comment: Line 152: It would be interesting to see an example

Answer: Considering the suggestions, we restructure the paragraph between lines 223 and 231.

Electrochemical methods employed on disease diagnosis

Comment: Line 206: A brief definition of an electroactive compound would improve this section of the discussion

Answer: It was described the meaning of an electroactive compound.

Comment: Table 1 should have “Measured / Controlled Property” rather than “Quantity”

Answer: Changes were performed.

Comment: Table 1 “Impedimetry” refers to Electrochemical impedance spectroscopy correct? If so the Controlled Property should be the frequency of potential oscillation in a given amplitude. As an example:

freq(Hz)=f(E±E_0 ),being E_0  the amplitude and E the average potential

Answer: The applied signal symbol was redefined.

Comment: Line 369: “small sinusoidal wave” is vague, instead the author may write: “through a sinusoidal wave of potential variation through time”

Answer: Indeed the sentence explaining the experimental conditions of electrochemical impedance techniques become more precise by describing the nature of the potential applied.

Comment: Line 360-377: this section lack a bit of continuity, the paragraphs cut harshly from subject to subject, I suggest that the authors reformulate each end and beginning of paragraph to make a continuous argument

Answer: This entire paragraph has been changed and improved as may be see in the paragraph  440-459.

Comment: Line 388: Spectroscopy

Answer: The change has been made. We kindly appreciate the error detection.

Comment: Line 390: Give an example of this sensing element

Answer: We appreciate the observation made and emphasize the fact that this example is mentioned in line 498 of the current document. “Recently, Donghai Lin et al. [68], have developed a simple and highly sensitive impedimetric immunosensor, based on the affinity reaction antibody-antigen establish on a paper electrode surface, for the sensitive detection of Escherichia coli O157:H7 bacteria.”

Comment: It would be interest to read about at least one example of pathology that may be surveyed with each of the electrochemical techniques: Potentiometry, amperometry, voltammetry and impedance spectroscopy. I suggest that the author write about a brief example of each case, in the pertinent paragraphs

Answer: In order to emphasize previously proclaimed sentence/paragraph, we also agree with the illustrative example exhibition about biological molecule potentiometric/amperometric detection. 

Reviewer 3 Report

In this article, the authors presented a systematic and comprehensive summary of biomolecules and electrochemical biosensors in chronic non-communicable disease surveillance. Electrochemical biosensors with excellent analytical performance represent a class of emerging biosensing technologies for many applications. The application focused on this article – chronic non-communicable disease surveillance – is highly significant. Therefore, the topic of this article fits perfectly with Biosensors. Notably, to the reviewer’s best knowledge, this topic has not been comprehensively reviewed in the literature so far. In the article, the authors started with the introduction of non-communicable diseases (NCDs), and then reviewed biosensors, especially electrochemical biosensors, employed in NCDs. The utilization of enzymes and nanoparticle catalysts in electrochemical biosensors were discussed in more detail. The authors also summarized the biomarkers used in NCDs, the reactive species (i.e., reactive oxygen species (ROS)) in the human body system, and ROS as signaling messengers of human health status. The relationship of ROS with NCDs was significantly discussed. The insights into electrochemical biosensor design and ROS bio-function summary should be of great interest to other researchers in the field. The perspectives provided in this article are believed to be able to inspire more excellent research work in the future. The reviewer would like to recommend the publication of this manuscript after following revisions are addressed:

  • In the introduction section, the authors should point out the features of this review, like why choose electrochemical biosensors for NCD surveillance? There many types of biosensing techniques, and what are the advantages of electrochemical biosensors over other types of biosensing techniques? Moreover, the authors should give a brief summary of this review, including what they are going to discuss, at the end of this section.
  • The authors are suggested to provide a figure to summarize the outline of the present review.
  • The contents in every section should be systematically and logically organized. From Sections 2 to 13, there are too many paragraphs in each section, and some of them only contain one sentence.
  • Please don’t use abbreviations in Keywords. “ROS” should be “reactive oxygen species”.
  • Page 19, Line 623, “Error! Reference source not found” and Page 40, Line 1229, “Error! Bookmark not defined” should be revised.

Author Response

Journal: Biosensors (ISSN 2079-6374)

Manuscript ID: Biosensors-897685

Type: Review

Title: Biomolecules and electrochemical tools in chronic non-communicable disease surveillance

Reviewer 3

Comment: In this article, the authors presented a systematic and comprehensive summary of biomolecules and electrochemical biosensors in chronic non-communicable disease surveillance. Electrochemical biosensors with excellent analytical performance represent a class of emerging biosensing technologies for many applications. The application focused on this article – chronic non-communicable disease surveillance – is highly significant. Therefore, the topic of this article fits perfectly with Biosensors. Notably, to the reviewer’s best knowledge, this topic has not been comprehensively reviewed in the literature so far. In the article, the authors started with the introduction of non-communicable diseases (NCDs), and then reviewed biosensors, especially electrochemical biosensors, employed in NCDs. The utilization of enzymes and nanoparticle catalysts in electrochemical biosensors were discussed in more detail. The authors also summarized the biomarkers used in NCDs, the reactive species (i.e., reactive oxygen species (ROS)) in the human body system, and ROS as signaling messengers of human health status. The relationship of ROS with NCDs was significantly discussed. The insights into electrochemical biosensor design and ROS bio-function summary should be of great interest to other researchers in the field. The perspectives provided in this article are believed to be able to inspire more excellent research work in the future. The reviewer would like to recommend the publication of this manuscript after following revisions are addressed:

In the introduction section, the authors should point out the features of this review, like why choose electrochemical biosensors for NCD surveillance? There many types of biosensing techniques, and what are the advantages of electrochemical biosensors over other types of biosensing techniques? Moreover, the authors should give a brief summary of this review, including what they are going to discuss, at the end of this section.

The authors are suggested to provide a figure to summarize the outline of the present review.

Answer: We appreciate excellent suggestions and corrections made.

Consequently, we have rearranged the interlude between lines 63 and 83 of the introduction section, in the current document.

“In the present study, the electrochemical techniques were selected for the surveillance of common NCD because it owns some special features that turn it one of the most suitable facilities for an early and accurate vigilance of health disorders among public health. Namely, their capacity to detect trace amounts of biological species, their miniaturisation that offers the possibility to have access to complex areas of the human organism, and their portability, which turns it easier to use in clinical applications.

In general, electrochemical appliances provide many advantages over conventional analytical system’s, like the ability to reproducibly handle minimal amounts of samples, low reagent consumption, reduced processing time, ease of proceeding and low-cost analysis.

The overwhelming increase in the number of chronic non-communicable diseases amongst the less elderly population is increasingly a concern worldwide. It is therefore essential to combat and prevent the progression of this kind of long-lasting health conditions, by defining monitoring strategies. These might be achieved through the applications of reliable and efficient techniques and methodologies that allow an accurate early detection, leading to proficient clinical outcomes that promote general public health.

Present work aims to review concepts related to most common NCD; biosensors; Electrochemical techniques employed on disease diagnosis; biological recognition elements for electrochemical sensing; Reactive species sources and role in physiological and pathological processes; Human body antioxidant mechanisms.

This overview sustains the development of tools with specific characteristics in order to trade them for use in biomolecule analysis and early detection of NCD.”

Comment: The contents in every section should be systematically and logically organized. From Sections 2 to 13, there are too many paragraphs in each section, and some of them only contain one sentence.

Answer: We appreciate the meticulous evaluation made in all sections.  

The current document has been restructured in order to proffer a more systematic and organized study.

Please don’t use abbreviations in Keywords. “ROS” should be “reactive oxygen species”.

Answer: We kindly appreciate for the perceptive repair. The abbreviation has been replaced by the extended designation “reactive oxygen species”.

Page 19, Line 623, “Error! Reference source not found” and Page 40, Line 1229, “Error! Bookmark not defined” should be revised.

Answer: We very much appreciate the observations and the disclosure of the accidental errors, which have already been corrected in the document.

Reviewer 4 Report

In the manuscript, the authors present a review regardind the importance of different biomolecules and their detection in the surveiilance in different chronic diseases. It is quite a very interesting subject, but, in my opinion this manuscript can not be published in this form.

  1. The manuscript is too long. Maybe is better to be published as a book chapter. 
  2. The authors did not present any review protocol.
  3. The authors did not present any eligibility criteria for the articles that were included in this review. Also, they did not present any methodology that was used when searching the literature (keywords, databases).
  4. The authors did not present any method for data extraction from different studies. 

Author Response

Journal: Biosensors (ISSN 2079-6374)

Manuscript ID: Biosensors-897685

Type: Review

Title: Biomolecules and electrochemical tools in chronic non-communicable disease surveillance

Reviewer 4

Comment: In the manuscript, the authors present a review regardind the importance of different biomolecules and their detection in the surveiilance in different chronic diseases. It is quite a very interesting subject, but, in my opinion this manuscript can not be published in this form.

The manuscript is too long. Maybe is better to be published as a book chapter.

Answer: We appreciate your observations about ours work subject.

Moreover, an excellent suggestion to consider the publication in the form in a book chapter.  However, we believe it is important to have the opportunity to publish current work in a magazine of major impact and reputation, as is the Biosensors journal.

Comment: The authors did not present any review protocol.

The authors did not present any eligibility criteria for the articles that were included in this review. Also, they did not present any methodology that was used when searching the literature (keywords, databases).

The authors did not present any method for data extraction from different studies.

Answer: We have followed the reviewer’s comments to solve this problem. Consequently, it was set-up a new session regarding PRISMA guidance used in literature survey, and the statistical analysis, following rejections and inclusions given the criteria applied in literature choice.

Round 2

Reviewer 4 Report

In the manuscript, the authors present a review regarding the importance of some biomolecules in the diagnosis of chronic non-communicable diseases. The manuscript is quite interesting. The manuscript has been reviewed before and the authors changed the manuscript according to the previous reviewers indications. Their comments are pertinent. That is why, I think that this manuscript can be published in this form.